# Identifying novel regulators of placental development using time-series transcriptome data

Ha TH Vu[1,2] , Haninder Kaur[1], Kelby R Kies[1,2], Rebekah R Starks[1,2], Geetu Tuteja[1,2]

**The placenta serves as a connection between the mother and the fetus during pregnancy, providing the fetus with oxygen, nutrients, and growth hormones. However, the regulatory mechanisms and dynamic gene interaction networks underlying early placental development are understudied. Here, we generated RNA-sequencing data from mouse fetal placenta at embryonic days 7.5, 8.5, and 9.5 to identify genes with timepoint-specific expression, then inferred gene interaction networks to analyze highly connected network modules. We determined that timepoint-specific gene network modules were associated with distinct developmental processes, and with similar expression profiles to specific human placental cell populations. From each module, we identified hub genes and their direct neighboring genes, which were predicted to govern placental functions. We confirmed that four novel candidate regulators identified through our analyses regulate cell migration in the HTR-8/SVneo cell line. Overall, we predicted several novel regulators of placental development expressed in specific placental cell types using network analysis of bulk RNA-sequencing data. Our findings and analysis approaches will be valuable for future studies investigating the transcriptional landscape of early development.**

## Introduction

The placenta is a transient organ that has critical roles during pregnancy, such as the transportation of oxygen and nutrients to the fetus, waste elimination, and the secretion of growth hormones. Placental defects are associated with devastating complications including preeclampsia and fetal growth restriction, which can lead to maternal or fetal mortality (1, 2). Therefore, it is fundamental to understand the mechanisms of placental development.

Because of ethical considerations and the opportunity for genetic manipulation, mouse models are frequently used when investigating early placental development. Like humans, mice have a hemochorial placenta (3), meaning that maternal blood directly comes in contact with the chorion. Although there are certain differences between the mouse and human placenta (3, 4), they do express common genes during gestation, including common regulators and signaling pathways involved in placental development (4, 5, 6, 7). For example, *Ascl2/ASCL2* and *Tfap2c/TFAP2C* are required for the trophoblast (TB) cell lineage in both mouse and human models (8, 9, 10). Another example is the HIF signaling pathway, which regulates TB differentiation in both mouse and human placenta (4).

Mouse placental development begins around embryonic day (e) 3.5 when the trophectoderm (TE) layer forms (5). The TE differentiates into different TB populations at e4.5, which eventually leads to the formation of the ectoplacental cone (EPC) (11). Between e7.5 and e9.5, the establishment of blood flow to the fetus begins, and highly dynamic changes in placental cell composition occur. At e7.5, the EPC is comprised of TB cells (3), organized into the inner and peripheral populations, with the inner cells actively proliferating and differentiating, whereas the outer cells can be invasive and interact with the decidua (11). Around e8.5, chorioallantoic attachment occurs, during which the chorion layer joins with the allantois (12). As a result, the e8.5 mouse fetal placenta includes cells from the EPC, chorion, and allantois (13). From e9.5 onward, the mouse fetal placenta is composed of distinct layers: the trophoblast giant cell (TGC) layer, the junctional zone (spongiotrophoblast and glycogen TB cells), and the labyrinth zone (chorion TB cells, syncytiotrophoblast [SCT] I and II cells, fetal endothelium, and spiral artery TGCs) (14, 15). Within the labyrinth layer, there is a dense network of vasculature where nutrients and oxygen are transported and exchanged. Although the structure of the placenta is not identical between mouse and human, certain mouse placental cell types are thought to be equivalent to human placental cell types (4). For example, parietal TGCs and glycogen TBs have been described as equivalent to human extravillous trophoblasts (EVTs) (4). Mouse TGCs are not as invasive as human EVTs (4), and they have different levels of polyploidy and copy-number variation (16); however, both EVTs and TGCs are able to degrade extracellular matrix to enable TB migration into the decidua (17).

---

[1]Genetics, Development, and Cell Biology, Iowa State University, Ames, IA, USA  [2]Bioinformatics and Computational Biology, Iowa State University, Ames, IA, USA

Correspondence: geetu@iastate.edu

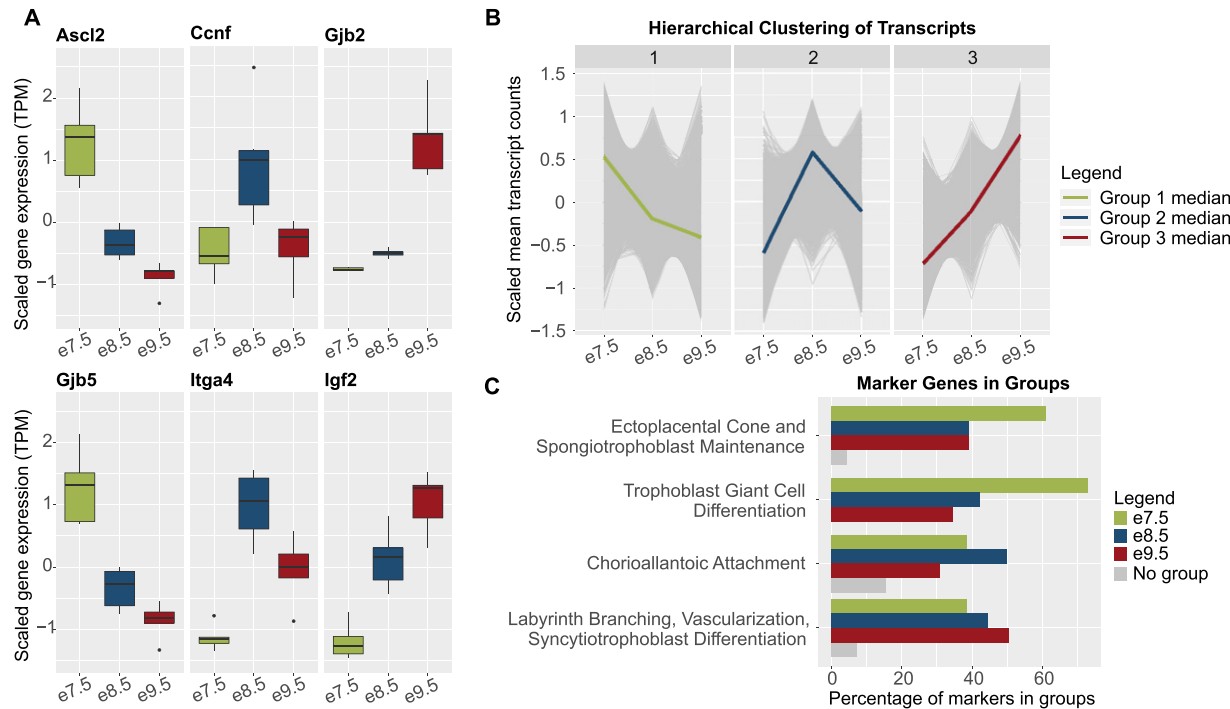

**Figure 1. Genes associated with distinct placental processes show timepoint-specific expression.**
**(A)** Boxplots of scaled mean expression (in transcripts per million, TPM) of marker genes showing timepoint-specific patterns. *Ascl2* and *Gjb5*, expected to peak at e7.5, markers of trophoblast proliferation and differentiation (8, 30); *Ccnf* and *Itga4*, expected to peak at e8.5, markers of chorioallantoic attachment (31, 32); and *Gjb2* and *Igf2*, expected to peak at e9.5, markers of nutrient transport (33, 34). **(B)** Line charts of scaled mean raw counts of transcripts in hierarchical clusters showing group median expression levels peak at each timepoint. **(C)** Bar plots showing that timepoint-associated hierarchical clusters captured most genes underlying distinct placental processes. Markers of timepoint-associated placental processes were obtained from previously published review articles (5, 18, 19, 20, 21). Green, markers in hierarchical cluster with a median expression level highest at e7.5; blue, markers in hierarchical cluster with a median expression level highest at e8.5; dark red, markers in hierarchical cluster with a median expression level highest at e9.5; and gray, markers in no hierarchical clusters.

Several individual regulators of the processes active between e7.5 and e9.5 have been identified, as reviewed in previously published articles (5, 18, 19, 20, 21). In addition, it is important to determine how these regulators potentially interact with other genes as networks. To identify novel regulators or infer gene interactions underlying developmental processes, unbiased whole-genome transcriptomic data can be used. Previous studies that used transcriptomics in the developing mouse placenta were either focused on analysis of one timepoint or focused on analysis of multiple -omics data (22, 23, 24, 25). Other studies of gene expression in human placenta across trimesters did not infer full gene interaction networks and instead focused on transcription factors (TFs) (16, 26). Single-cell analysis has been used to investigate cell type–specific gene expression in the placenta; however, these studies do not predict regulators underlying specific placental development processes (27, 28).

Here, we generated RNA-sequencing (RNA-seq) data from mouse fetal placental tissues at e7.5, e8.5, and e9.5. We then carried out clustering, differential expression, and network analyses to infer gene interactions and predict novel regulators of placental development. We further demonstrated that our network constructions could be used to infer cell populations in the mouse placenta at the three timepoints. Finally, we conducted in vitro validation experiments and confirmed that

several genes we identified have a role in regulating TB cell migration.

# Results

## Genes associated with distinct placental processes show timepoint-specific expression

We generated and analyzed transcriptomic data from fetal placental tissues at e7.5, e8.5, and e9.5 to identify genes regulating distinct processes during placental development. Based on the stages of placental development and the cell types present at each stage, we predicted that genes with highest expression at e7.5 would be involved in TB proliferation or differentiation; genes with highest expression at e8.5 would have a role in chorioallantoic attachment; and genes with highest expression at e9.5 would have a role in the establishment of nutrient transport. Indeed, we observed that previously identified regulators of TB proliferation and differentiation (e.g., ASCL2 (8, 29), GJB5 (30)), chorioallantoic attachment (e.g., CCNF (31), ITGA4 (32)), and nutrient transport (e.g., GJB2 (33), IGF2 (34)) showed timepoint-specific patterns that matched with our predictions (Fig 1A). Next, we performed hierarchical clustering to determine whether protein-coding transcripts

would cluster into groups that displayed timepoint-specific expression. From this analysis, we obtained three groups of transcripts in which the median expression was highest at e7.5 (8,242 transcripts, equivalent to 5,566 genes), e8.5 (8,091 transcripts, equivalent to 5,536 genes), and e9.5 (7,238 transcripts, equivalent to 5,347 genes) (Fig 1B and Table S1). Hereafter, these groups are referred to as hierarchical clusters.

To evaluate the computational robustness and biological significance of the hierarchical clusters, we carried out additional analyses. First, we used three different algorithms: K-means clustering, self-organizing maps, and spectral clustering, to validate the trends of the expression levels in hierarchical groups, and the number of transcript groups ($k$ = 3, 4, and 5). Only with $k$ = 3 did we obtain groups with median expression-level trends consistent in all four algorithms (Fig S1). Moreover, with $k$ = 3, the maximum percent of agreement (see the Materials and Methods section) between hierarchical clusters and clusters obtained using each of the different algorithms was 70.34–87.26% (Fig S1), whereas the maximum percent of agreement between hierarchical clusters and clusters obtained from other algorithms decreases to between 55.67 and 65.72% with $k$ = 4, and between 54.81 and 59.19% with $k$ = 5.

Second, we compared our hierarchical group data with previously published mouse and human placental microarray time course data from Soncin et al (7). Despite the technical differences between the datasets, we observed a consensus that our e7.5 hierarchical cluster had the highest percent of overlap compared with the Soncin et al gene groups that are down-regulated over time, and that our e9.5 hierarchical cluster had the highest percent of overlap compared with the Soncin et al gene groups that either have highest expression at e9.5–e12.5 or are up-regulated over time (Table S1).

Lastly, we determined how the genes in each cluster relate to processes of placental development. From previously published review articles (5, 18, 19, 20, 21), we acquired gene sets associated with distinct processes, namely, EPC and/or *spongiotrophoblast maintenance* (expected to be most active at e7.5, when the EPC is still in a highly proliferative state (11)); TGC *differentiation* (expected to be more active at e7.5 because the mouse placenta at e8.5 and e9.5 includes more differentiated TB subtypes (3, 13, 14, 15)); *chorioallantoic attachment* (expected to be most active at e8.5 (12)); and *labyrinth branching, vascularization,* and *SCT differentiation* (expected to be most active at e9.5, after these processes have initiated (21)) (Table S1). Indeed, we observed that the e7.5 hierarchical cluster captured the most genes in the *EPC* and *spongiotrophoblast maintenance* and *TGC differentiation* groups; the e8.5 hierarchical cluster included the most genes in the *chorioallantoic attachment* group; and the e9.5 hierarchical cluster included the most genes in the *labyrinth branching, vascularization,* and *SCT differentiation* group (Fig 1C and Table S1). Together, these data demonstrate that hierarchical clustering can be used to obtain transcript groups that are associated with relevant biological processes at each timepoint, but is not sufficient to fully distinguish processes that may have varied activity levels throughout time.

To this end, and because hierarchical clustering is sensitive to small perturbations in the datasets (35), we carried out differential expression analysis (DEA) and identified transcripts and genes with the strongest changes over time (Fig S2 and Table S2). After combining results from hierarchical clustering and DEA, we defined

timepoint-specific gene groups (see the Materials and Methods section for gene group definitions; Fig S2) and obtained 922 e7.5-specific genes, 915 e8.5-specific genes, and 1952 e9.5-specific genes (Table S3). Gene ontology (GO) analysis showed that the timepoint-specific gene groups were enriched for highly relevant biological processes such as "TGC differentiation" (e7.5-specific genes), "labyrinthine layer development" (e8.5- and e9.5-specific genes), "blood vessel development" (e7.5- and e9.5-specific genes), and "response to nutrient" (e9.5-specific genes) (Table S3).

It is possible that timepoint-specific groups share genes that have timepoint-specific transcripts. Indeed, we identified 37 genes shared between e7.5 and e8.5, five genes shared between e7.5 and e9.5, and 109 genes shared between e8.5 and e9.5 (Table S3). We found that genes only present at one timepoint (timepoint-unique genes) were generally enriched for similar terms as the full group of timepoint-specific genes (Table S3). However, terms related to the development of the labyrinth layer such as "labyrinthine layer morphogenesis" and "labyrinthine layer blood vessel development" were only enriched when using all e8.5-specific genes but not when using e8.5 timepoint-unique genes. Moreover, we found that, unlike genes shared between e9.5 and e7.5, genes shared between e9.5 and e8.5 were enriched for processes such as "blood vessel development" and "insulin receptor signaling pathway." This observation may indicate that different transcripts of the same genes could be expressed at different timepoints for the continuation of certain biological processes.

### Network analysis reveals potential regulators of developmental processes in the placenta

To predict interactions among timepoint-specific genes and subset timepoint-specific genes into regulatory modules, we used the STRING database (36) and GENIE3 (37) (see the Materials and Methods section). With the two approaches of network inference, we were able to predict networks of genes by means of previously published experimental results and text-mining of available publications (STRING), and de novo computational analysis with random forest–based methods (GENIE3). We then carried out network subclustering with the GLay algorithm (38) (see the Materials and Methods section) and identified four network modules at e7.5, six at e8.5, and eight at e9.5 (Table S4 and Fig S3). To determine whether the networks were associated with distinct processes of placental development, we used GO enrichment analysis.

Compared with e8.5 and e9.5 networks, e7.5 networks had a higher rank or fold change and were significantly enriched for the GO terms "inflammatory response" (e7.5_1_STRING: $-\log_{10}$[q-value] = 22.82 and e7.5_2_GENIE3: $-\log_{10}$[q-value] = 3.95) and "female pregnancy" (e7.5_2_GENIE3: $-\log_{10}$[q-value] = 4.1) (Fig 2A and Table S5). The term "morphogenesis of a branching structure," which can be expected after chorioallantoic attachment around e8.5, was not enriched at e7.5, but was enriched in multiple e8.5 and e9.5 networks (e8.5_1_STRING: $-\log_{10}$[q-value] = 1.73, e8.5_2_GENIE3: $-\log_{10}$[q-value] = 1.72, e9.5_1_STRING: $-\log_{10}$[q-value] = 4.01, e9.5_1_GENIE3: $-\log_{10}$[q-value] = 1.54, e9.5_2_STRING: $-\log_{10}$[q-value] = 14.33, and e9.5_2_GENIE3: $-\log_{10}$[q-value] = 2.2). After chorioallantoic attachment is complete, nutrient transport is being established. Accordingly, we observed the following enrichments:

**Table 1. Hub genes associated with each network.**

| Timepoint | Network | Number of hub genes | Hub genes |
|---|---|---|---|
| e7.5 | e7.5_1_STRING | 7 | ***Mmp9***, *Ptprc*, *Tlr2*, *Cd68*, *Ctss*, *Cybb*, *Itgb2* |
| | e7.5_2_GENIE3 | 10 | ***Nr2f2*** (40), ***Hmox1*** (41), *Prdm1* (42), *Ctbp2* (43), *Irx1*, **Frk**, *Siah2*, *Satb1*, *Trim8*, *Irf9* |
| e8.5 | e8.5_1_STRING | 11 | ***Akt1*** (44), ***Mapk14*** (45), *Mapk1* (46), *Adam10*, *Creb1*, *Apob*, *Cdh2*, *Cttn*, **Hsp90aa1**, *Apoe*, *Casp3* |
| | e8.5_2_GENIE3 | 17 | *Erf* (47), *Setd2* (48), **Msx2**, **Dvl3**, *Dnmt1*, *Dnmt3b*, *Lin28a*, *Chek1*, *Dnajc2*, *Vgll1*, *Gpbp1l1*, *Jade1*, *Myef2*, *Nfxl1*, *Rbmx*, *Rhox4e*, *Cenpf* |
| e9.5 | e9.5_1_STRING | 7 | ***Fbxl19***, ***Smurf1***, ***Ubc***, *Wnt5a*, **Ube2d1**, **Mgrn1**, **Nedd4** |
| | e9.5_1_GENIE3 | 34 | ***Rb1*** (49), ***Yap1*** (50), *Esx1* (51), ***Ncoa3***, ***Ski***, ***Pitx1***, ***Zfx***, *Peg3*, *Ash1l*, **Arid1b**, **Arrb1**, **Prmt2**, **Tulp1**, **Vgll4**, **Creg1**, **Foxo3**, **Hif1an**, *Apbb1*, *2700081O15Rik*, *Ankrd2*, *Bbx*, *Cdk5*, *Hdac6*, *Mllt3*, *Calcoco1*, *Cavin1*, *Cenpb*, *Cited4*, *Dtx1*, *Fam129b*, *Hcfc2*, *Mlxip*, *Phf8*, *Tsc22d1* |
| | e9.5_2_STRING | 15 | ***Cdh1*** (52), ***Fn1*** (53), ***Igf2*** (54), ***Tgfb1*** (55), ***Vegfa*** (56), *Egfr* (57), ***Col1a1***, ***Csf1***, ***Timp1***, *Igf1*, **App**, **Spp1**, **Itpkb**, **Qsox1**, *Gas6* |
| | e9.5_2_GENIE3 | 27 | ***E2f8*** (58), ***Vegfa*** (56), ***Tead2*** (59), ***Ets1***, ***Orc2***, ***Sox18***, *Klf3*, *Mrtfb*, *Trip6*, **Cbx7**, **Prnp**, **Arhgef5**, **Pias1**, **Pias3**, **Rasd1**, **Txnip**, **Zfp362**, *Plagl1*, *5730507C01Rik*, *BC004004*, *Bhlhe40*, *Ctdsp1*, *Grhl1*, *Ell2*, *Phf2*, *Fam83g*, *Rhox12* |
| | e9.5_3_STRING | 5 | ***Gaa***, **Lpcat1**, **Olr1**, **Cd59a**, *Stom* |
| | e9.5_3_GENIE3 | 29 | ***Tead1*** (59), ***Smad4*** (60), ***Smad5*** (61), *Foxf1* (62), ***Cebpb***, ***Jun***, ***Hes1***, ***Tead3***, ***Cbx4***, ***Cdk2***, ***Dapk3***, ***Gata1***, *Hsbp1*, *Ncor2*, **Dpf3**, **Limd1**, **Loxl2**, **Pcgf5**, *Elmsan1*, *Hexim1*, *Lhx2*, *Sin3b*, *Mta3*, *Jade1*, *Rbm15b*, *Rhox4g*, *Rhox6*, *Rhox9*, *Tcf25* |
| | e9.5_4_STRING | 10 | *Lpar3* (63), ***Gna12***, ***Gnas***, **Acta2**, **Gcgr**, **Pik3r3**, **Rhoc**, **Rhog**, *Rhoj*, *Adcy4* |

Colored genes are ones that have annotated or possible roles in placental development (see the Materials and Methods section); green, e7.5-specific genes; blue, e8.5-specific genes; and brown, e9.5-specific genes. Genes in bold are hub genes in one network inference method and nodes in the other method's networks.

"endothelial cell proliferation" (highest ranked in e9.5_2_STRING: $-\log_{10}$[q-value] = 15.91); "lipid biosynthetic process" (only significant after e7.5, highest ranked in e9.5_3_STRING: $-\log_{10}$[q-value] = 17.63); "cholesterol metabolic process" (only significant after e7.5, highest ranked in e9.5_2_GENIE3: $-\log_{10}$[q-value] = 2.76 and e9.5_3_STRING: $-\log_{10}$[q-value] = 7.79); and "response to insulin" (only significant after e7.5, highest ranked in e9.5_1_GENIE3: $-\log_{10}$[q-value] = 1.67). "Placental development," "vasculature development," and cell migration–related terms ("positive regulation of cell migration" and "epithelium migration") are observed in networks at all timepoints, although these terms are not consistently ranked in all networks. Using randomization tests, we observed most of these GO terms (10 of 11 terms) were significantly enriched when using the network genes but not random gene sets (significance level of 0.05; the term "vasculature development" having $P$-value = 0.0549 and 0.0575 with subnetworks e9.5_1_GENIE3 and e9.5_3_GENIE3, respectively) (see the Materials and Methods section; Fig S3). This analysis demonstrates that the network genes were highly relevant to the biological functions of interest. Moreover, the observed GO terms strongly aligned with the processes enriched when using the full lists of timepoint-specific genes (Table S3), indicating the representative characteristics of the network genes. Although the current analysis focuses on the biological processes related to placental

development, there are other terms significantly enriched, which can be found in Table S5. In summary, we identified 18 subnetworks across three timepoints for downstream analyses, some of which were enriched, according to GO analysis and randomization tests, for specific terms relating to placental development (Fig 2A).

We predicted that hub genes, defined to be nodes with high degree, closeness, and shortest path betweenness centrality in the networks (see the Materials and Methods section), could be potential regulators of developmental processes in the placenta. We first determined whether the hub genes from each network with enriched GO terms described in the previous paragraph were directly annotated or possibly related to placental functions using the Mouse Genome Informatics (MGI) database (39) (see the Materials and Methods section; Tables 1 and S6). Briefly, genes annotated under any GO or MGI phenotype terms related to placenta, TB cells, TE, and the chorion layer are considered as having a "known" role in the placenta. Genes annotated under terms related to embryo are considered as having a "possible" role in the placenta, because embryonic lethal mouse knockout lines frequently have placentation defects, and because defects in placental development can be associated with the development of other embryonic tissues (64, 65, 66). Hereafter, such genes are referred to as "known/possible genes". In the e7.5 networks, there were 17 hub

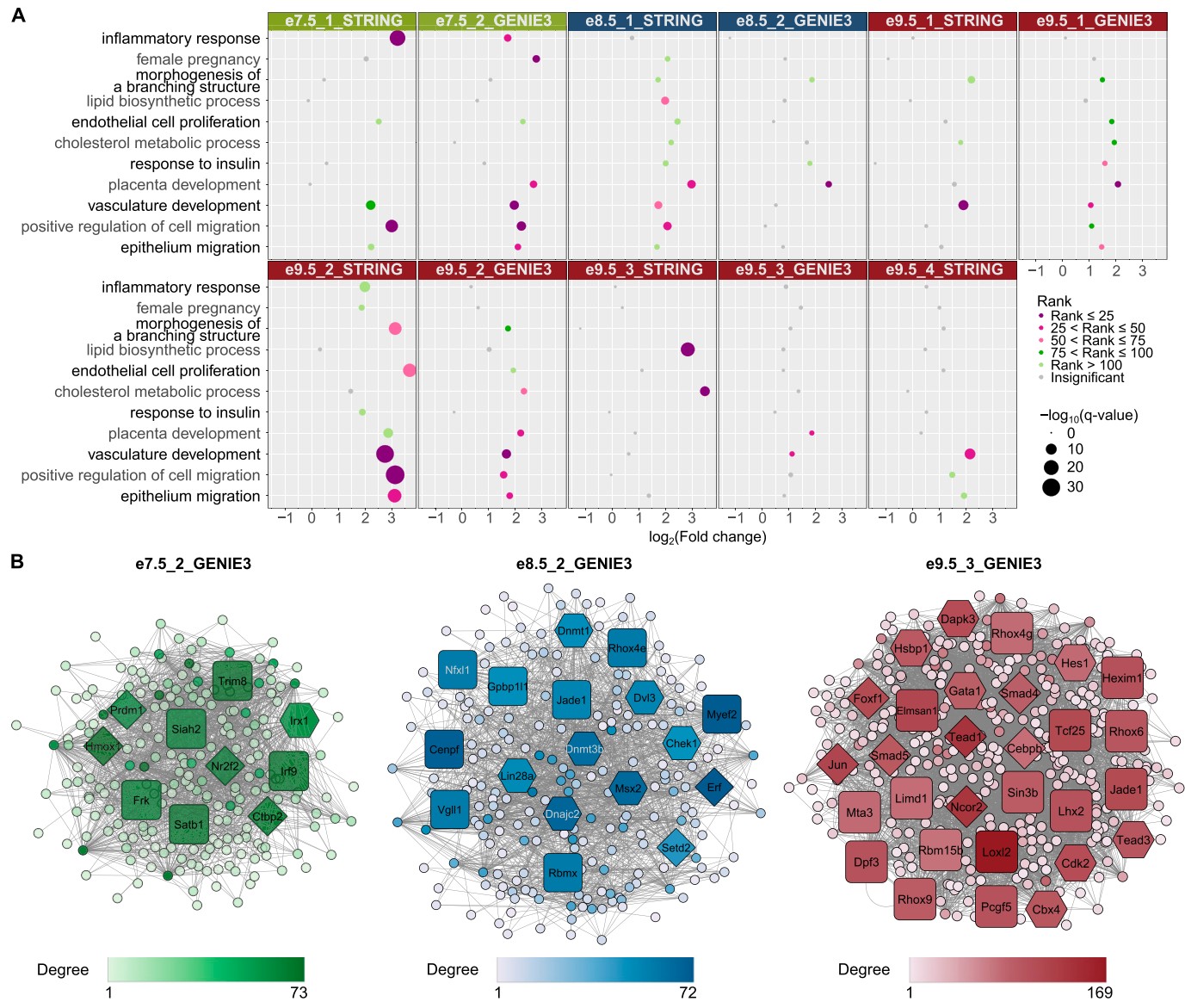

**Figure 2. Network analysis identifies gene modules with relevant functions and reveals potential regulators of placental development.**
**(A)** Gene ontology (GO) analysis of networks demonstrates the association of gene sets with placental development processes. Only selected terms are shown. Dot colors correspond to ranks of the terms in each analysis; dot sizes correspond to –log$_{10}$(q-value). A GO term is considered enriched if its q-value ≤ 0.05, fold change ≥ 2, and the number of observed genes ≥ 5. For full GO enrichment analysis, see Table S5. **(B)** Network analysis highlights potential regulators of placental development. **(A)** Only a subset of networks with enriched terms from (A) are shown. Diamond shape, hub genes with known roles in placenta; hexagon, hub genes with possible roles in placenta; and rounded square, hubs without related annotation. Color: the darker the color is, the higher the node's degree centrality is. For visualization of all other subnetworks, see Fig S3.

genes in which six genes were known/possible. The number of hub genes that are labeled as known/possible is statistically significant when comparing to random gene sets selected from the e7.5 timepoint-specific group (Fig S3). In the e8.5 and e9.5 networks, 17 of 28 and 48 of 127 hub genes were known/possible, respectively. Similar to e7.5, the number of hub genes labeled as known/possible in both e8.5 and e9.5 networks was statistically significant when comparing to random gene sets selected from the corresponding timepoint-specific groups (Fig S3). These results indicate that the gene sets we identified are significantly associated with relevant phenotypes in the mouse.

In the network e7.5_1_STRING, we identified seven hub genes (Table 1), one of which (*Mmp9*) was considered as possibly related to placental development according to the MGI database. Although *Mmp9* was not annotated directly to placenta using our definition of a placenta term on the MGI database, it has been shown to be required for proper implantation, TB differentiation, and invasion (67). In the network e7.5_2_GENIE3 (Fig 2B), 10 hub genes were identified, five of which were annotated as regulating or having possible roles in placental development in the MGI database. Four of the genes are required for TB proliferation, differentiation, migration, or invasion, namely, NR2F2 (40), PRDM1 (42), HMOX1 (41), and

CTBP2 (43) (Tables 1 and S6). Other hub genes could be novel regulators of placental functions. One example is *Frk*, a hub gene of the e7.5_2_GENIE3 network, which has been suggested to inhibit cell migration and invasion in human glioma (68) and retinal carcinoma cells (69), but has not been studied in early placental development. Although the networks inferred by the two methods did not share any hub genes, hub genes identified with one method could be members of the other method's networks. These hub genes are *Mmp9* (e7.5_1_STRING), *Frk*, *Hmox1*, and *Nr2f2* (e7.5_2_GENIE3) (Table 1). This observation strengthens the potential roles of the *Frk* gene in placental development.

At e8.5, hub genes included both novel and known genes in placental development or chorioallantoic attachment. For example, in the network e8.5_1_STRING, 11 hub genes were identified, eight of which were known/possibly associated with placental development according to the MGI database. For example, the genes AKT1, MAPK1, and MAPK14 have a role in placental vascularization (44, 45, 46) (Tables 1 and S6). For the network e8.5_2_GENIE3 (Fig 2B), there were 17 hub genes identified (Tables 1 and S6), with nine genes having known or possible functions. E8.5_2_GENIE3 network's hub genes include genes that are not known/possible, but have been studied in the context of placental development such as *VGLL1* (7). Hub genes identified with one method and present in the other method's networks are *Hsp90aa1*, *Akt1*, and *Mapk14* (e8.5_1_STRING), and *Dvl3* and *Msx2* (e8.5_2_GENIE3) (Table 1). An example of a novel gene is *Jade1* (hub node of e8.5_2_GENIE3), which has been found to have high expression in extraembryonic ectoderm and TB cells and hence may play roles in placental vascularization by interacting with VHL (70), but has not been tested functionally in placental tissues.

From the e9.5 networks, we identified 127 hub genes of which 48 have known/possible functions in placental development in the MGI database (Tables 1 and S6). For instance, in e9.5_1_GENIE3, e9.5_2_STRING, and e9.5_4_STRING, hub genes that regulate labyrinth layer development include EGFR (57) and RB1 (49), and hub genes that regulate placental vasculature development include FN1 (53) and VEGFA (71). Hub genes identified with one method and present in the other method's networks include important genes such as *Rb1* (49), *Yap1* (50) (e9.5_1_GENIE3), and *Vegfa* (e9.5_2_STRING) (Table 1). Notably, *Vegfa* is the only hub gene identified with both network inference methods. There are also hub genes known to be important for placental nutrient transport such as *Igf2* (34), and other genes that could be novel regulators. For example, LHX2 is part of the mTOR signaling pathway in osteosarcoma (72), but has yet to be studied in placenta although the mTOR signaling pathway is known to be involved in nutrient transport in the placenta (73).

In summary, we have identified hub genes in networks at each timepoint. Analyzing the annotations of hub genes using the MGI database demonstrated that the hub genes are biologically relevant to mouse development and will be strong candidates for future investigation.

### Timepoint-specific genes can be associated with cell-specific expression profiles of human placenta

To determine whether timepoint-specific genes could capture different placental cell populations, we carried out deconvolution analysis with LinSeed (74) and inferred the cell-type profiles. Briefly, LinSeed takes advantage of the mutual linearity relationships between cell-specific genes and their corresponding cells to infer the topological structures underlying cell populations of tissues. This approach would enable us to use bulk RNA-seq data to predict proportions of cell types in the mouse placenta without prior knowledge of cell-type markers or matching single-cell datasets. As input to LinSeed, we used the 5,000 most highly expressed genes across all timepoints (expression in TPM), from which 1,413 genes were found to be statistically significant for the inference models and thus used to conduct the deconvolution analysis (see the Materials and Methods section; Fig S4). As a result, we observed five cell groups, which captured 99% of the variance in the placenta tissue samples (Fig S4). Among these groups, e7.5 samples had the highest proportion of cell group 3, e8.5 samples had the highest proportion of cell group 2, and e9.5 samples had the highest proportion of cell group 5 (Fig 3A, left panel; Table S7). Cell group 1 and cell group 4 did not have consistent cell proportions across biological replicates of a single timepoint. The identification of these cell groups could have resulted from noise introduced by both biological variation and technical variation, which is challenging to overcome when using a small sample size or analyzing without prior knowledge of the deconvolution analysis. Therefore, we focused on cell groups 3, 2, and 5. We identified 100 markers (see the Materials and Methods section) for cell group 3, 100 markers for cell group 2, and 41 markers for cell group 5. Interestingly, 95 of the 100 markers of cell group 3 are e7.5-specific genes, 45 of the 100 markers of cell group 2 are e8.5-specific genes, and 40 of the 41 markers of cell group 5 are e9.5-specific genes (Fig 3A, right panel; Table S7). This indicates that the independent timepoint-specific gene analysis we performed in the previous section could represent gene profiles of distinct cell populations.

To this end, we used the PlacentaCellEnrich web tool to annotate timepoint-specific genes with human placental cell types (75). At all timepoints, we observed enrichment suggesting the presence of TB cells. Specifically, the e7.5-specific genes not only were most significantly enriched for genes with EVT-specific expression ($\log_2$[fold] = 1.75, $-\log_{10}$[adj. *P*-value] = 4.18), but also had enrichment for SCT ($\log_2$[fold] = 1.1, $-\log_{10}$[adj. *P*-value] = 2.09), the e8.5-specific group was only enriched for genes that had villous cytotrophoblast (VCT)–specific expression ($\log_2$[fold] = 1.51, $-\log_{10}$[adj. *P*-value] = 2.36), and the e9.5-specific group had the highest enrichment for genes with fetal fibroblast–specific expression ($\log_2$[fold] = 2.04, $-\log_{10}$[adj. *P*-value] = 22.04) (Figs 3B and S5). We note that the e9.5-specific group had enrichment for genes with cell type–specific expression in multiple cells, including endothelial cells ($\log_2$[fold] = 2.02, $-\log_{10}$[adj. *P*-value] = 18.66), VCT ($\log_2$[fold] = 1.5, $-\log_{10}$[adj. *P*-value] = 7.38), SCT ($\log_2$[fold] = 1.23, $-\log_{10}$[adj. *P*-value] = 6.93), and EVT ($\log_2$[fold] = 1.05, $-\log_{10}$[adj. *P*-value] = 3.05) (Figs 3B and S5). Together, this demonstrates that our analysis is picking up on the diverse cell populations present at e9.5 compared with e7.5.

Motivated by the fact that cell-specific expression profiles for multiple human placental cell types are enriched at e7.5 and e9.5, we hypothesized that the gene network modules at each timepoint could capture specific cell populations. Indeed, PlacentaCellEnrich analysis of e7.5_2_GENIE3 network genes was significantly enriched for genes with EVT-specific expression ($\log_2$[fold] = 2.32, $-\log_{10}$[adj.

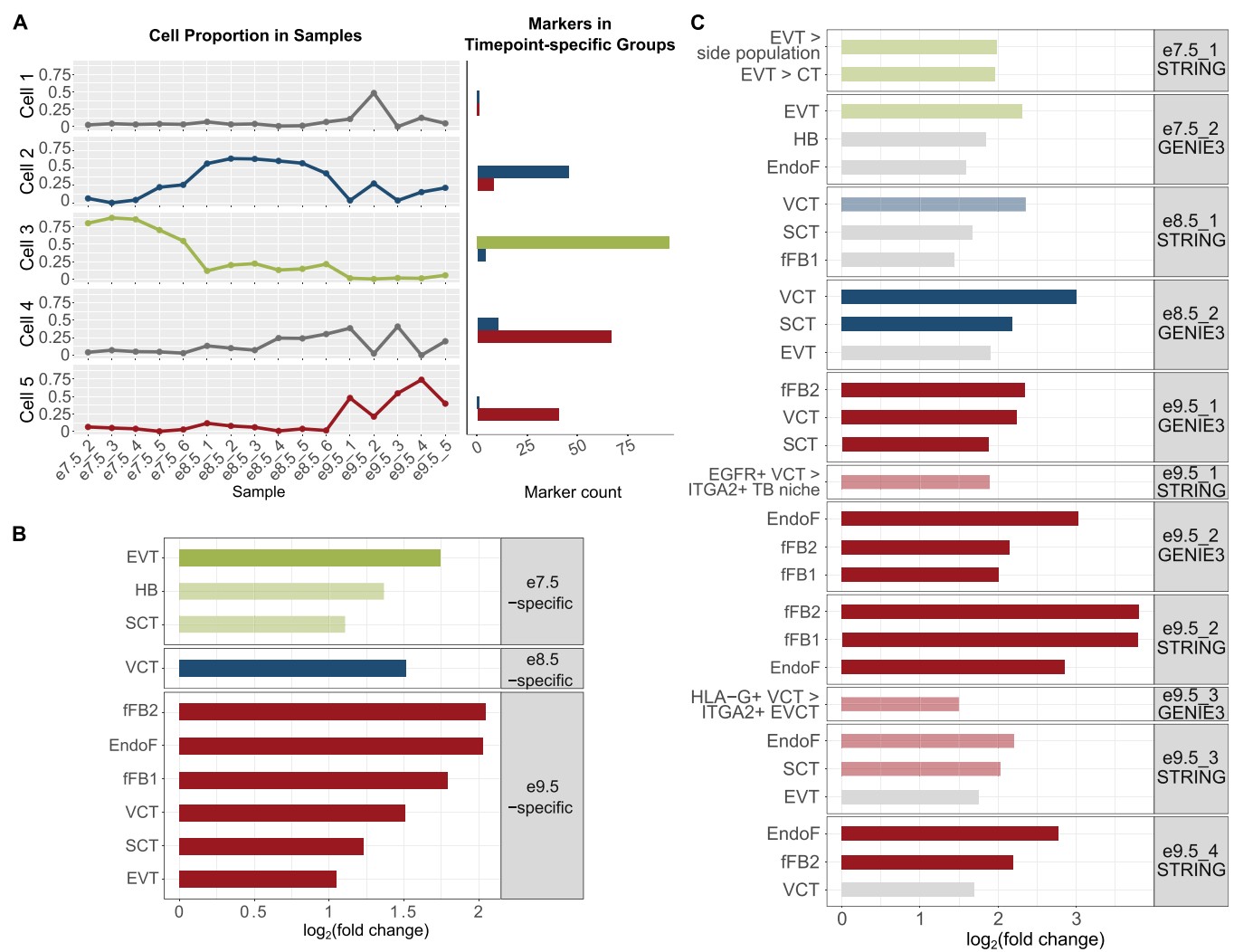

**Figure 3. Timepoint-specific gene groups can be associated with human placenta cell–specific expression profiles.**
**(A)** Deconvolution analysis using LinSeed showed five cell groups, three of which had highest proportions in e7.5 samples (group 3), e8.5 samples (group 2), and e9.5 samples (group 5). Also using LinSeed, we identified markers of each cell group and observed a high number of genes in common with timepoint-specific genes (cell group 3 with e7.5-specific genes, cell group 2 with e8.5-specific genes, and cell group 5 with e9.5-specific genes). Left panel: line charts showing cell proportions in each sample; right panel: bar plots showing the number of cell markers in each timepoint-specific gene group. **(B)** Bar plots showing that timepoint-specific genes share similar profiles to these of human placental cell populations. Enrichment analysis was carried out with PlacentaCellEnrich using first trimester human placenta single-cell RNA-seq data to determine gene groups with cell type–specific expression. A significant enrichment has adj. *P*-value ≤ 0.05, fold change ≥ 2, and the number of observed genes ≥ 5. The lightness of the colors corresponds to adj. *P*-value; the lighter colors, 0.005 < adj. *P*-value ≤ 0.05; and the darker colors, adj. *P*-value ≤ 0.005. Only enrichments for cells of fetal origin are shown. Full enrichment results (including both maternal and fetal cells) are shown in Fig S5. **(C)** Bar plots showing that network genes share similar profiles of specific human placental cell populations. **(B)** Enrichment analysis was carried out with PlacentaCellEnrich as in (B) and placenta ontology. Gray, adj. *P*-value > 0.05; the lighter colors, 0.005 < adj. *P*-value ≤ 0.05; and the darker colors, adj. *P*-value ≤ 0.005. For PlacentaCellEnrich, three fetal cell types with the lowest adj. *P*-values are shown. For placenta ontology, selected enrichments are shown. Full enrichment results (including both maternal and fetal cells and for every network) of PlacentaCellEnrich are shown in Fig S5. Full enrichment results (for every network) of placenta ontology are in Table S8. Abbreviations: SCT, syncytiotrophoblast; HB, Hofbauer cells; EVT, extravillous trophoblast; VCT, villous cytotrophoblast; EndoF, fetal endothelium; fFB1, fetal fibroblast cluster 1; fFB2, fetal fibroblast cluster 2; EVT > side population, GSE57834_extravillous_trophoblast_UP_side_population (genes up-regulated in EVT compared with side population—original data from GSE57834); EVT > CT, GSE57834_extravillous_trophoblast_UP_cytotrophoblast (genes up-regulated in EVT compared with cytotrophoblast—original data from GSE57834); EGFR+ VCT > ITGA2+ TB niche, GSE106852_EGFR+_UP_ITGA2+ (genes up-regulated in EGFR+ villous cytotrophoblast compared with ITGA2+ proliferative trophoblast niche—original data from GSE106852); and EGFR+ VCT > HLA-G+ EVCT, GSE80996_EGFR+_villous_cytotrophoblast_UP_HLA_G+_proximal_column_extravillous_cytotrophoblast (genes up-regulated in EGFR+ villous cytotrophoblast compared with HLA-G+ proximal column extravillous cytotrophoblast, original data from GSE80996).

*P*-value] = 1.67) (Figs 3C and S5), but no longer with genes that have SCT-specific expression. Both E8.5_1_STRING and e8.5_2_GENIE3 were enriched for genes with VCT-specific expression (log$_2$[fold] = 2.35 and 3, −log$_{10}$[adj. *P*-value] = 1.43 and 5.41, respectively). In addition to VCT-specific expression, e8.5_2_GENIE3 had enrichment

for genes that had SCT-specific expression (log$_2$[fold] = 2.19, −log$_{10}$ [adj. *P*-value] = 2.93) (Figs 3C and S5). At e9.5, genes in the networks e9.5_1_GENIE3 and e9.5_3_STRING showed strong enrichment for TB-specific expression, such as in SCT and VCT. On the contrary, e9.5_2_GENIE3, e9.5_2_STRING, e9.5_3_GENIE3, and e9.5_4_STRING had

strong enrichment for fetal fibroblast and endothelium–specific expression profiles (Figs 3C and S5). Importantly, randomization tests showed that the enrichment of cell type–specific genes was only significant in these subnetworks but not in random gene sets selected from corresponding timepoint-hierarchical groups (Fig S6), which highlights the biological relevance of the gene network modules.

For genes in networks e7.5_1_STRING, e9.5_1_STRING, and e9.5_3_GENIE3, we did not observe any enrichment for fetal placental cells, possibly because not all genes in the networks are annotated in the first trimester dataset (28) used when calculating cell enrichments in PlacentaCellEnrich. Therefore, we also used placenta ontology (76), which carries out enrichment tests based on different datasets than those used in PlacentaCellEnrich. With e7.5_1_STRING, in agreement with previous analyses on e7.5-specific genes or genes in the e7.5_2_GENIE3 network, we observed annotations related to EVT cells being enriched, such as "EVT > side population" ($log_2$[fold] = 1.99 and false discovery rate [FDR] = 0.027) and "EVT > CT" ($log_2$[fold] = 1.96 and FDR = 0.028) (Table S8). With e9.5_1_STRING, the term "EGFR+ VCT > ITGA2+ TB niche" was enriched ($log_2$[fold] = 1.89 and FDR = 0.023), meaning there are a significant number of genes in this network that were up-regulated in EGFR+ VCT compared with the ITGA2+ proliferative TB niche in the first trimester placenta. Similarly, with e9.5_3_GENIE3, we found the term "EGFR+ VCT > HLA-G+ EVCT" enriched ($log_2$[fold] = 1.5 and FDR = 0.043), which means there are a significant number of genes in this network that were up-regulated in EGFR+ VCT compared with HGL-A+ proximal column extravillous cytotrophoblast in the first trimester placenta. In the other networks, placenta ontology enrichment results generally agreed with PlacentaCellEnrich (Table S8). Together, the PlacentaCellEnrich and placenta ontology analyses provide evidence that network analysis can be used to identify genes more likely associated with specific placental cell types.

In summary, we have demonstrated that the identification of timepoint-specific gene groups and densely connected network modules can be used to infer the cellular composition of bulk RNA-seq samples. We used independent human datasets from different sources to annotate the cell types in each timepoint's sample. As a result, from the bulk RNA-seq data we were able to observe that at e7.5 and e8.5, there were a high proportion of different TB populations, whereas at e9.5, the placental tissues consisted of multiple cell types such as TB, endothelial, and fibroblast cells.

### Gene knockdown provides further evidence for a role of network genes in the placenta

As described in the Network analysis reveals potential regulators of developmental processes in the placenta section, we identified hub nodes, and as a result also obtained genes directly connected to the hub nodes (Table S6). Many of the genes (23 genes at e7.5 and 208 genes at e9.5) had drastic expression changes over time (having at least one transcript with fold change ≥ 5 between e7.5 and e9.5) (Table S9), which may be more likely to have regulatory roles specific to processes or cell types associated with each timepoint. However, there were several hub genes and genes directly connected to the hub nodes that were differentially expressed (DE) but had lower fold changes and showed high expression across all timepoints. We predict these highly expressed genes to be generally important for TB

function and processes such as cell migration, a term that was associated with multiple timepoint-specific networks (Fig 2A).

To investigate this further, we performed gene knockdown and migration assays for four candidate genes from four different networks in the HTR-8/SVneo cell line, an established model for studying TB migration (55, 77, 78). From the lists of hub genes and their directly connected nodes (Table S6), we obtained genes that met the following criteria: having expression levels > 5 TPM in the mouse placenta transcriptome data we generated, having expression levels > 5 FPKM (Fragments Per Kilobase of transcript per Million of mapped reads) in human TB cell lines (79) and having expression levels > 20 TPM in the HTR-8/SVneo cell line (23) (Table S6). From this list, we selected four genes: *Mtdh* and *Siah2* (from the e7.5_1_STRING and e7.5_2_GENIE3 network, respectively), *Hnrnpk* (from the e8.5_2_GENIE3), and *Ncor2* (from the e9.5_3_GENIE3), all of which were nodes in networks annotated as TB subtypes (see the Timepoint-specific genes can be associated with cell-specific expression profiles of human placenta section).

For each of the four genes, we transfected two different siRNAs, and all eight siRNAs resulted in high knockdown efficiencies (74–93%; Fig 4A). Each pair of siRNAs similarly reduced target protein levels (Fig S7). Next, we performed cell migration assays and visually observed a reduction in cell migration capacity for all four genes (Figs 4B and S7). To determine whether the observed reduction in cell migration was statistically significant, we further quantified the integrated cell densities (Fig 4C and Table S10). For *SIAH2* and *HNRNPK*, integrated densities of cells were significantly decreased upon knockdown with both siRNAs using a *P*-value ≤ 0.05. Specifically, for *SIAH2*, the densities reduced by 98.57% ± 0.42% (mean ± SE) and 83.87% ± 12.1% with siRNA #1 and siRNA #2, respectively. For *HNRNPK*, the densities reduced by 99.55% ± 0.09% with siRNA #1 and 98.68% ± 0.2% with siRNA #2. For *MTDH* and *NCOR2*, the reductions were significant for one siRNA (*MTDH*, siRNA #2, 98.55% ± 0.86%; *NCOR2*, siRNA #1, 98.11% ± 0.09%), and were fair for the other siRNA, possibly because of the variable results between biological replicates (*MTDH*, siRNA #1, 55.28% ± 17.22%; *NCOR2*, siRNA #2, 81.27% ± 14.04%). When comparing the number of cells 48 h post-transfection for cells treated with target gene siRNA to cells treated with negative control siRNA, we determined that none of the target gene siRNA treatments resulted in significant changes in cell counts. We do note that *SIAH2* siRNA #1 has some decrease in cell counts (*P*-value = 0.081), and *NCOR2* siRNA #1 and *NCOR2* siRNA #2 have some increase in cell counts (*P*-value = 0.081 and *P*-value = 0.077) compared with negative control–treated samples (Fig S7). This provides evidence that, in general, the reduction in cell migration capacity was likely not due to the target gene impacting the rate of cell death. Overall, these results confirm that network analysis and gene filtering based on defined criteria can identify genes important for TB function.

## Discussion

Placental development involves multiple processes that are active during different stages of gestation. Using transcriptomic data generated from mouse placenta at e7.5, e8.5, and e9.5, we identified timepoint-

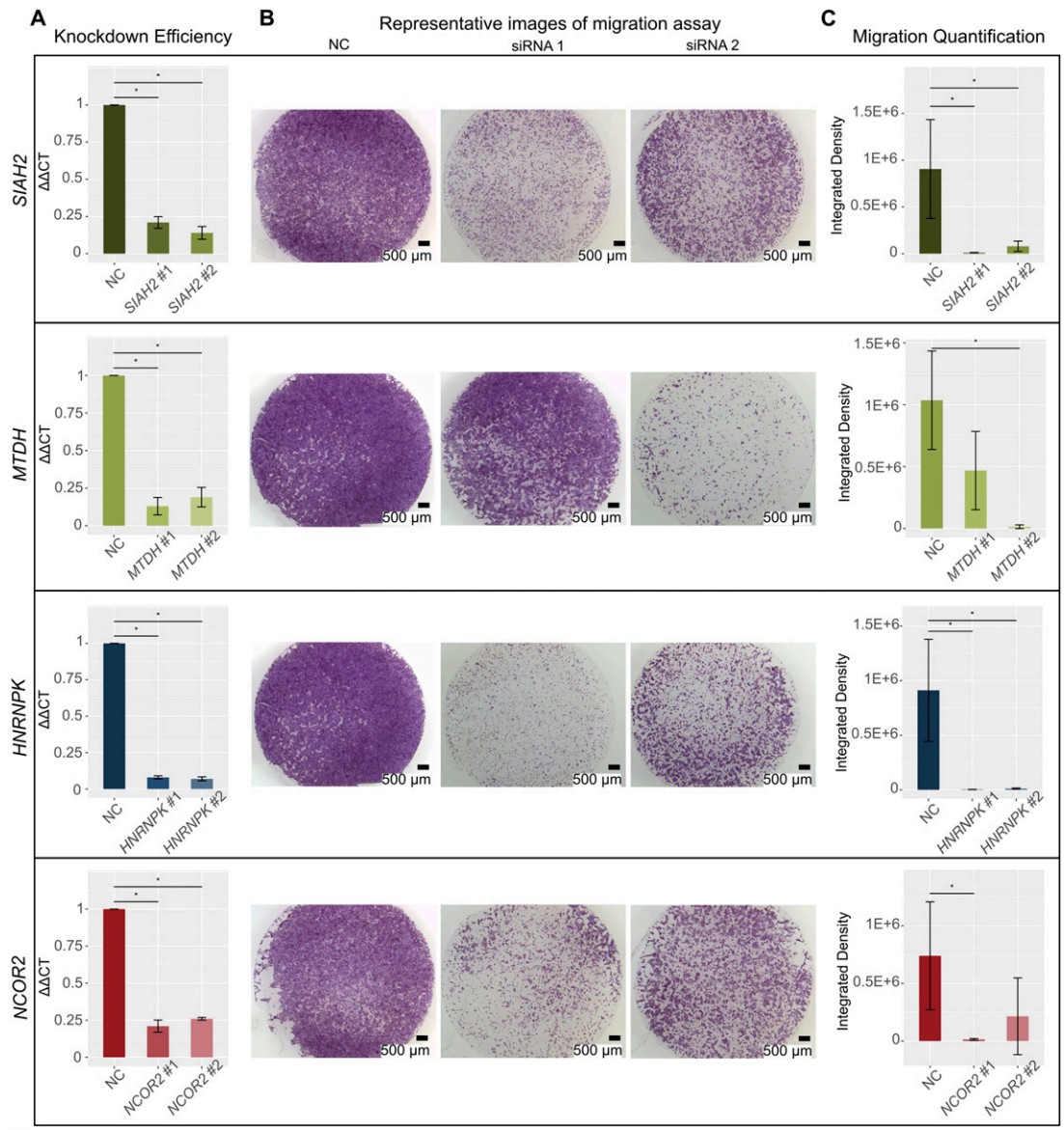

**Figure 4. Gene knockdown (KD) of selected network genes showing reduction in cell migration capacity.**
Panels correspond to four genes, *SIAH2, MTDH, HNRNPK,* and *NCOR2*. Each condition, negative control (NC), siRNA #1, and siRNA #2, had three biological replicates. Error bars show SD. **(A)** Bar plots showing that gene expression was significantly reduced after KD compared with NC. *GAPDH* was used for normalization of all four genes' expression (ΔCT). Percent KD was calculated with the ΔΔCT method. Values shown were normalized to the NC siRNAs. The y-axis shows ΔΔCT value. Details of KD efficiencies, siRNAs, and primer sequences can be found in Tables S10 and S11. (*) indicates *P*-value < 0.05 (one-sided Wilcoxon rank sum test, n = 3). **(B)** Representative images of migration assays. Left, NC samples; middle, siRNA #1 samples; right, siRNA #2 samples. Scale bar: 500 μm. **(C)** Bar plots showing significant reduction in the integrated density of cells after KD compared with NC samples. The y-axis shows integrated densities of cells in NC samples, samples KD with siRNA #1 of each gene, and samples KD with siRNA #2 of each gene. Details of integrated densities can be found in Table S10. (*) indicates *P*-value < 0.05 (one-sided Wilcoxon rank sum test, n = 3).

specific gene groups that can be used for gene network inferences and analyses, and for cell population annotations. Importantly, we were able to infer cell populations at different timepoints without known marker genes or reference datasets from the same species. The cell proportion inferences were necessary to bypass the confounding factors from cell heterogeneity, and thus predict more accurate novel regulators of cell-specific processes such as TB cell migration. This computational pipeline could be used to infer and analyze gene networks governing the development of placenta at other timepoints or to study developmental processes in other tissues.

Combining hierarchical clustering with DEA, we were able to identify gene groups using an unsupervised approach. It has also been shown that for times-series analyses with fewer than eight timepoints, pairwise DEA combined with additional methods identifies a more robust set of genes (80). Alternatively, model-based clustering using RNA-seq profiles (81) could also be useful for gene group identification. However, it is still important to evaluate the robustness and functional relevance of the fitted models by carrying out additional downstream analyses.

We carried out DEA across the three timepoints on both the transcript and the gene level. These analyses revealed that a gene may have transcripts that are DE at different timepoints. For example, IGF2, a placental nutrient transport marker (82), has different transcripts grouped to e8.5 and e9.5 (Table S1). This observation also aligned with a recent study, which showed in 6–10 and 11–23 wk of human placenta, DE genes, transcripts, or differential transcript usage could all assist in the understanding of placental development (83). Therefore, in future studies, investigating roles of both genes and their transcripts could give a more complete functional profile at each timepoint. Moreover, our results, together with previous studies in human placenta (16, 26, 83), suggest that time-series transcriptomic analyses could be a useful approach to identify genes governing the development of the placenta. It will be beneficial to integrate these time-series datasets to determine species-specific biomarkers of placental development.

We identified hub genes and their immediate neighboring genes, which could regulate placental development and confirmed the roles of four novel genes in regulating cell migration in the HTR-8/SVneo cell line. These genes were selected primarily based on the network analyses, but also based on expression data from human cells to account for possible differences between mouse and human placental gene expression. Previous studies suggested these four candidates are functionally important in mouse. MTDH has been suggested to regulate cell proliferation in mouse fetal development (84). The *Siah* gene family is important for several functions (85). Of relevance to the placenta, SIAH2 is an important regulator of HIF1α during hypoxia both in vitro and in vivo (86). Moreover, although *Siah2* null mice exhibited normal phenotypes, combined knockouts of *Siah2* and *Siah1a* showed enhanced lethality rates, suggesting the two genes have overlapping modulating roles (87). *Hnrnpk*$^{-/-}$ mice were embryonic lethal, and *Hnrnpk*$^{+/-}$ mice had dysfunctions in neonatal survival and development (88). *Ncor2*$^{-/-}$ mice were embryonic lethal before e16.5 because of heart defects (89). According to the International Mouse Phenotyping Consortium database (90), *Ncor2* null mice also showed abnormal placental morphology at e15.5. However, none of these genes have been studied in the context of TB migration. We observed that although all siRNAs were able to decrease cell migration capacity, there was variability in the amount of decrease, even when comparing two siRNAs targeting the same gene. This observation did not seem to be associated with differences in transcript or protein knockdown levels and could be due to different off-target effects for different siRNAs. Moreover, we observed that cell counts generally were not decreased upon target gene knockdown compared with negative control knockdown. However, more detailed analysis and process-specific assays are needed. For example, future studies assessing each gene's role in cell adhesion, cell–cell fusion, cell proliferation, and cell apoptosis can be done to better understand their roles in placental development. We also acknowledge the HTR-8/SVneo cell line bears certain differences to TB cells such as in their miRNA expression profiles (91). Therefore, to determine the exact roles of these genes in the placenta, future experiments in human TB stem cells derived with the Okae protocol (79) or gene knockout experiments in vivo are necessary.

Interestingly, all four genes have been shown to have roles in cancer cells: SIAH2 was shown to promote cell invasiveness in human gastric cancer cells by interacting with ETS2 and TWIST1 (92); MTDH regulates proliferation and migration of esophageal squamous cell carcinoma cells (93); the absence of HNRNPK reduces cell proliferation, migration, and invasion ability in human gastric cancer cells (94); and repression of NCOR2 and ZBTB7A increased cell migration in lung adenocarcinoma cells (95). This result further supports previous studies that show the comparability between placental cell migration and invasion, and tumor cell migration and invasion (76, 96), although specific genes may have different impacts on migration/invasion capacity such as with the *NCOR2* gene.

In our analyses, we observed that timepoint-specific genes and their networks represented expression profiles for specific placental cell populations at the three timepoints. In particular, analysis of e7.5-specific and e8.5-specific genes and networks showed that placental tissues at e7.5 and e8.5 contain different populations of TB cells, whereas e9.5-specific genes and networks showed multiple cell types including TB, endothelial, and fibroblast cells. The significant overlap between e7.5-specific genes and genes of EVT cells yielded an interesting suggestion that the TB cell populations in e7.5 mouse placenta may share similarity in gene profiles to human EVT, although mouse TB and human EVT have certain differences such as their invasiveness levels (4) and levels of polyploidy and copy-number variation (16). Examples of EVT genes present in the e7.5-specific gene group include *FSTL*3 (down-regulation decreased TB migration and invasion in the JAR cell line (97)), *ADM* (increased TB migration and invasion in the JAR and the HTR-8/SVneo cell line (98)), and ASCL2 (regulates TB differentiation (8)). Moreover, hub genes could be used to identify potential novel markers for the cell types corresponding to their subnetworks. For example, hub genes of subnetworks enriched for SCT-specific genes such as *Dvl3* (e8.5_2_GENIE3) and *Olr1* (e9.5_3_STRING) are not established SCT marker genes, but are in fact DE in SCT compared with human trophoblast stem cells, EVT (99) or endovascular TB (100). In general, combining network analysis with existing gene expression data from single-cell or pure-cell populations will allow identification of novel cell-specific marker genes to help future studies focused on different TB populations.

Although it is true that data at single-cell (sc) resolution are necessary to gain more insight into cell populations in heterogeneous tissues, these results showed strong evidence that bulk RNA-seq data could be used to infer the cell-type composition. In addition, scRNA-seq assays could be noisier than bulk RNA-seq because of various technical aspects such as the amount of starting materials, cell size, cell cycle, and batch effects (101, 102), which are difficult to correct (103). Therefore, bulk RNA-seq, ideally in conjunction with scRNA-seq, is beneficial for the study of biological processes that involve multiple cell types. Nevertheless, we acknowledge that our deconvolution analysis and cell-type annotations were limited because of the absence of matching scRNA-seq data, data from pure-cell populations, or extensive cell marker lists. As these types of information become more available, deconvolution analysis can be used to identify species-specific cell types or correct for confounding effects before DEA (104).

In our network analysis, we observed that the GO term "inflammatory response" was enriched in e7.5_1_STRING (q-value = $1.52 \times 10^{-23}$), e7.5_2_GENIE3 (q-value = 0.00012), and e9.5_2_STRING (q-value = $4.17 \times 10^{-10}$) (Table S5). The inflammatory process could be happening in the placenta during e7.5 to e9.5 when TB cells actively invade the decidua (65) and create a pro-inflammatory environment (105). Another possibility is contamination from decidual cells, which could be detected when combining bulk and scRNA-seq (106). This further demonstrates the benefits of bulk and scRNA-seq data integration.

Upon conclusion of this study, we have shown that in the mouse placenta at e7.5, e8.5, and e9.5, genes with timepoint-specific expression patterns can be associated with distinct processes and cell types. The genes identified by timepoint-specific gene network analysis could be interesting candidates for future studies focused on the understanding of placental development and placenta-associated pregnancy disorders.

# Materials and Methods

### RNA-seq library preparation and sequencing

Placental tissue was collected from timed-pregnant CD-1 mice (Charles River Labs) following the guidelines and protocol approved by Iowa State University Institutional Animal Care and Use Committee, protocol number 18–350. Placenta samples were collected as previously described (22, 107) at e7.5, e8.5, and e9.5, and the age of the embryo was determined by following the embryonic development guidelines (108). Briefly, tissues from the EPC and chorion were separated from the decidua, yolk sac, umbilical cord, and embryo, and then collected. For e7.5, 12 EPCs were collected and pooled into one replicate, as described in (24). For e8.5, five placentas were collected per replicate, and for e9.5, one placenta was collected per replicate. Each timepoint had a total of six biological replicates.

Tissues were processed for RNA isolation immediately after collection using the PureLink RNA micro scale kit (12183016; Thermo Fisher Scientific). RNA concentration and RIN values were measured using the RNA 6000 Nano assay kit on the Agilent 2100 Bioanalyzer (GTF Facility, ISU), and all samples had a RIN score ≥ 7.7 (Table S11). Further processing of the samples, and library preparation and sequencing were performed by the DNA facility at Iowa State University. Libraries were sequenced using the Illumina HiSeq 3000 with single-end 50 base pair reads. The pooled library sample was run over two sequencing lanes (technical replicates for each sample).

### RNA-seq data processing

The quality and adapter content were assessed using FastQC (version 0.11.7) (109). Low-quality reads and adapters were trimmed with Trimmomatic (version 0.39) (110).

Technical replicates were then merged, and the reads were pseudo-aligned and quantified (in TPM) using Kallisto (version 0.43.1; $l$ = 200, $s$ = 30, $b$ = 100) (111). Transcript sequences on

autosomal and sex chromosomes of the mouse genome (GRCm38.p6) from Ensembl release 98 (112) were used to build the Kallisto index.

For further quality control, we carried out hierarchical clustering and principal component analysis of samples. First, from the transcripts with raw counts ≥ 20 in ≥ 6 samples, we obtained the top 50% most variable transcripts, then centered and scaled their expression. Next, we implemented hierarchical clustering with the hclust() function in R (package *stats* (113), version 3.6.3), using the agglomerative approach with the Euclidean distance and complete linkage. To implement principal component analysis, we used the prcomp() function in R (package *stats*, version 3.6.3). We observed samples of each timepoint cluster close to each other and away from other timepoints. Outlier samples, which did not cluster with their respective timepoint groups, were removed before carrying out downstream analyses (Fig S8).

### Cluster analysis

Before performing all clustering procedures, transcripts with low raw counts (mean raw counts < 20 in all timepoints) were filtered out, and expression data (in TPM) were scaled and recentered. Hierarchical clustering, k-means clustering, self-organizing map, and spectral clustering were performed on the top 75% of most variable protein-coding transcripts (23,571 transcripts in total).

We implemented hierarchical clustering with the hclust() function in R (package *stats* (113), version 3.6.3), using the agglomerative approach with the Euclidean distance and complete linkage. The resulting dendrogram was cut at the second highest level to obtain three clusters. To test the robustness of the clustering assignments, we also carried out clustering with the number of clusters as 4 and 5.

K-means clustering was carried out using the R function kmeans() (*centers* = 3, 4, and 5; other parameters: default; package *stats*, version 3.6.3).

Self-organizing map clustering was performed with the R function som() with rectangular 3 × 1, 4 × 1, and 5 × 1 grid (other parameters: default; package *kohonen* (114), version 3.0.10).

To implement spectral clustering, we used the following functions in R: computeGaussianSimilarity() ($\Sigma$ = 1) to compute similarity matrix, and spectralClustering() ($K$ = 3, 4, and 5; other parameters: default; package RclusTool (115), version 0.91.3) to cluster.

The percent agreement between cluster assignments of different methods was quantified as (number of transcripts in common between two clusters)/(total number of transcripts in two clusters) × 100.

To determine how the genes in each cluster relate to specific processes of placental development, we obtained gene lists from previously published review articles (5, 18, 19, 20, 21), then calculated the percentage of markers in hierarchical clusters as (number of markers in a cluster)/(total number of markers of the process) × 100.

### DEA

DEA at transcript and gene levels was carried out with Sleuth (version 0.30.0) (116) using the likelihood-ratio test (default basic

filtering) and the *P*-value aggregation process (117). Fold change of a transcript was calculated using its average raw TPM across all samples. A transcript was considered DE if it had a fold change ≥ 1.5 and a q-value ≤ 0.05. A gene was considered DE if its q-value was ≤ 0.05 and had at least one protein-coding DE transcript. For lists of DE protein-coding transcripts that had at least one DE gene, and lists of DE genes with at least one DE protein-coding transcripts, see Table S2.

## Definition of timepoint-specific genes

Timepoint-specific gene groups are defined as the following:

(i) e8.5-specific transcripts: transcripts in e8.5 hierarchical cluster, are up-regulated at e8.5 (compared with e7.5), or are up-regulated at e8.5 (compared with e9.5). E8.5-specific genes are ones associated with e8.5-specific transcripts.
(ii) e7.5-specific transcripts: transcripts in e7.5 hierarchical cluster, are up-regulated at e7.5 (compared with e9.5), and are not in the e8.5-specific group. E7.5-specific genes are ones associated with e7.5-specific transcripts.
(iii) e9.5-specific transcripts: transcripts in e9.5 hierarchical cluster, are up-regulated at e9.5 (compared with e7.5), and are not in the e8.5-specific group. E9.5-specific genes are ones associated with e9.5-specific transcripts.

## Network construction and analysis

The STRING database (version 11.0b) (36) was used to build protein–protein interaction networks at each timepoint. Edges from evidence channels: experiments, databases, text-mining, and co-expression with a confidence score ≥ 0.55, were chosen for further analyses.

Gene regulatory networks at each timepoint were constructed with GENIE3 (version 1.16.0) (37). At each timepoint, as inputs for GENIE3, timepoint-specific transcripts with average TPM at the timepoint ≥ 5 were aggregated to obtain gene counts with the R package tximport (version 1.14.2; *countsFromAbundance* = length-ScaledTPM) (118). Genes that encode TFs and co-TFs, downloaded from AnimalTFDB (version 3.0) (119), were treated as candidate regulators. Then, edges with weight < the $90^{th}$ percentile were filtered out.

Largest connected components of the networks were analyzed using Cytoscape (version 3.7.2) (120). All networks were treated as undirected, and network subclustering was performed using the GLay plug-in (default parameters) (38). Networks with ≥ 100 nodes were used for further analyses. Hub genes were defined as nodes that have degree, betweenness, and closeness centralities in the $10^{th}$ percentile of their networks.

A gene was determined to have an annotated role in placental development if it was annotated under all GO and MGI phenotype terms related to placenta, TB cells, TE, and chorion layer. A gene was categorized as having possible roles in placental development if it was annotated under all GO and MGI phenotype terms related to embryo. GO terms, MGI phenotype terms, and gene annotations were downloaded from MGI (http://www.informatics.jax.org/) (version 6.19) (39). For lists of terms used, see Table S6.

Randomization tests were carried out to determine whether the number of known/possible hub genes at a timepoint is significant. For each timepoint, from the respective timepoint-specific groups, 10,000 gene sets of the same number as the hub gene numbers were sampled. Then, the number of known/possible genes in each set was counted. A *P*-value was calculated as the number of times a random gene set has ≥ known/possible genes than the observed number, divided by 10,000.

## GO analyses

To determine the relevant functions of the gene lists, we used GO analysis. clusterProfiler (version 4.0.5) (121) was used, with the mouse annotation from the org.Mm.eg.db R package (version 3.13.0) (122), the maximum size of genes = 1,000, and a q-value cut-off = 0.05. Next, a fold change for each term was calculated as GeneRatio/BgRatio. A GO term was considered enriched when its q-value ≤ 0.05, fold change ≥ 2, and the number of observed genes ≥ 5. Hypergeometric test was used for enrichment following the suggestions in Rivals et al (123).

Randomization analysis was carried out to determine whether a GO term is statistically significant for a subnetwork's genes. For each subnetwork, from the respective timepoint-hierarchical groups, 10,000 gene sets with the same size as the subnetwork were sampled. For each of the random sets, the q-value of a specific term with ClusterProfiler (same settings as above) was obtained. Then, the *P*-value of the randomization test was calculated as the number of random gene sets with q-values lower than the q-value of that term in the original subnetwork, divided by 10,000.

## Deconvolution analysis

To infer the proportion of cell types across timepoints, we carried out deconvolution analysis using the R package LinSeed (version 0.99.2) (74). Gene abundance (in TPM) used as inputs for the analysis was obtained using tximport (version 1.14.2; *countsFromAbundance* = lengthScaledTPM) (118). Then, we used the top 5,000 most expressed genes across timepoints, and sampled 100,000 times to test for the significance of the genes to be used for deconvolution analysis. A significant gene was one with *P*-value ≤ 0.05. The number of cell groups was determined after examining the singular value decomposition plot, generated with the svdPlot() function in LinSeed. Cell markers were defined as the top 100 genes closest to the cell group's corner, and closer to the corner than any other corners.

## Placenta cell enrichment and placenta ontology analyses

The PlacentaCellEnrich web tool (75) and placenta ontology (76) were used to infer the relevant cell types using gene lists. For PlacentaCellEnrich, cell type–specific groups were based on single-cell transcriptome data of the first trimester human maternal–fetal interface from Vento-Tormo et al (28). An enrichment was considered significant if its adj. *P*-value is ≤ 0.05, fold change ≥ 2, and the number of associated genes found is ≥ 5. For placenta ontology, we obtained placenta ontology GMT file from Naismith et al and uploaded the file to the WEB-based GEne SeT AnaLysis Toolkit (www.webgestalt.org) (124) as a functional database. An ontology

with FDR ≤ 0.05, fold change ≥ 2, and the number of observed genes ≥ 5 was considered enriched. To avoid duplication while sampling, only genes with one-to-one pairwise orthology were considered for the enrichment tests.

Randomization analysis was carried out to determine whether the enrichment of human first trimester placenta cell type–specific genes is statistically significant for a subnetwork's gene. For each subnetwork, from the respective timepoint-hierarchical groups, 10,000 gene sets with the same size as the subnetwork were sampled. For each of the random sets, the adjusted $P$-value of a specific cell-type enrichment with PlacentaCellEnrich (same settings as above) was obtained. Then, the $P$-value of the randomization test was calculated as the number of random gene sets with adjusted $P$-values lower than the adjusted $P$-value of that cell type in the original subnetwork, divided by 10,000.

## In vitro validation experiments

### Cell culture

HTR-8/SVneo (CRL3271; ATCC) were cultured as recommended by ATCC and as done by others ([125]). Briefly, cells were grown in RPMI-1640 media (302001; ATCC) supplemented with 5% FBS (97068-085; VWR) without antibiotics. Cells were split every 3–4 d, at 80–90% confluency.

### siRNA knockdown

HTR-8/SVneo cells were transfected with two different siRNAs for each target gene knockdown (KD). Cells were split at 80% confluency, and siRNA transfection was performed in six-well plates; 150,000 cells/well were seeded ([23]). After 24 h, cells were transfected with 30 nM siRNA using RNAiMAX 3000 (13778150; Thermo Fisher Scientific). Media were replaced after 24 h of transfection, and then, cells were collected after 48 h of transfection, counted using the TC20 Automated Cell Counter (Bio-Rad), and seeded for migration assays. The remaining cells were pelleted to isolate RNA using the Invitrogen RNA mini kit (12183018A; Thermo Fisher Scientific). The RNA concentration was determined using a NanoDrop, and 200 ng of the RNA was reverse-transcribed to cDNA (4368814; Thermo Fisher Scientific). KD efficiencies were checked by qRT-PCR using primers listed in Table S11. *GAPDH* was used for normalization of all four genes' expression (ΔCT). Percent KD was calculated with the ΔΔCT method. siRNA and primer information can be found in Table S11.

### Migration assays

Migration assays were performed using Costar inserts (3464; Corning). The inserts were placed in a 24-well plate, and 75,000 cells in serum-free RPMI media (30-2001; ATCC) were directly seeded in the top chamber of the insert. The bottom chamber was filled with 600 µl of RPMI media supplemented with 10% FBS as a chemoattractant. The cells were allowed to migrate for 24 h at 37°C. The cells on the bottom of the inserts were fixed in 4% PFA (AAJ61899AK; Thermo Fisher Scientific) for 5 min and then washed for 1 min with PBS twice. The cells in the top chamber were scraped off using a wet Q-tip (22029488; Thermo Fisher Scientific), and the cells on the bottom of the inserts were stained with hematoxylin (23245677; Thermo Fisher Scientific) for 24 h. The inserts were washed twice in distilled water. The membrane was cut using a scalpel (1484002; Thermo Fisher Scientific) and mounted on a clean glass slide in VectaMount Mounting Medium (NC9354983; Thermo Fisher Scientific). The cells were observed under a dissection microscope and imaged at 12.5× magnification. The images were analyzed using the ImageJ tool, and the integrated density was obtained for each image.

### Western blot

After siRNA KD, whole-cell lysate (4× Laemmli protein sample buffer, 1610747; Bio-Rad) or cytoplasmic extract (NE-PER Extraction Kit, 78833; Thermo Fisher Scientific) was resolved using SDS–PAGE and transferred to the nitrocellulose membrane (1620113; Bio-Rad) using the Trans-Blot Turbo transfer system (1704150; Bio-Rad). After protein transfer, membranes were blocked and probed with antibodies as listed in Table S11.

### Statistical analysis

Experiments were performed with three replicates per condition (negative control or knockdown) per gene. $P$-values were calculated with the one-sided Wilcoxon rank sum test to test for a significant decrease in cell migration, and the two-sided Wilcoxon rank sum test for cell count comparisons.

## Data Availability

All code for the analyses is available at https://github.com/Tuteja-Lab/PlacentaRNA-seq. All raw and processed data are available for download on NCBI Gene Expression Omnibus Repository, accession number: GSE202243.

## Supplementary Information

## Acknowledgements

We acknowledge the Iowa State University DNA Facility for preparing and sequencing the RNA-seq libraries, and the Research IT group at Iowa State University (http://researchit.las.iastate.edu) for providing servers and IT support. We would like to thank Tuteja laboratory members for their discussion and support. This work was supported in part by the Eunice Kennedy Shriver National Institute of Child Health & Human Development of the National Institutes of Health under award number R01HD096083 (to G Tuteja). G Tuteja is Pew Scholar in the Biomedical Sciences, supported by The Pew Charitable Trusts. The views expressed are those of the author(s) and do not necessarily reflect the views of the funding agencies.

### Author Contributions

HTH Vu: conceptualization, formal analysis, investigation, methodology, and writing—original draft, review, and editing.
H Kaur: validation, investigation, methodology, and writing—review and editing.
KR Kies: validation and writing—review and editing.

RR Starks: methodology and writing—review and editing.
G Tuteja: conceptualization, supervision, funding acquisition, investigation, methodology, and writing—original draft, review, and editing.

## Conflict of Interest Statement

The authors declare that they have no conflict of interest.

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
