## [Reviewer comments · Life Science Alliance]

Life Science Alliance

Identifying novel regulators of placental development using time-series transcriptome data

Ha Vu, Haninder Kaur, Kelby Kies, Rebekah Starks, and Geetu Tuteja

DOI: <https://doi.org/10.26508/lsa.202201788>

Corresponding author(s): Geetu Tuteja, Iowa State University

Review Timeline:

Submission Date:	2022-10-26
Editorial Decision:	2022-11-17
Revision Received:	2022-11-21
Accepted:	2022-11-22

Transaction Report:

Please note that the manuscript was reviewed at Review Commons and these reports were taken into account in the decision-making process at Life Science Alliance.

Manuscript number: RC-2022-01491

Corresponding author(s): Geetu Tuteja

1. General Statements [optional]

We thank all of the reviewers for seeing the value in our work, both from the perspective of the analysis approaches, which could be applied to other aspects of development, as well as our specific findings as they relate to placental development. We greatly appreciate all of the reviewer comments and suggestions, and our responses, including details of manuscripts edits, are below. We believe that incorporating the reviewer suggestions has strengthened our manuscript and it is now suitable for publication.

This section is mandatory. Please insert a point-by-point reply describing the revisions that were already carried out and included in the transferred manuscript.

Reviewer #1:

Review of "Identifying novel regulators of placental development using time series transcriptomic data and network analyses."

The authors present a detailed bioinformatic assessment of mouse developmental time series of the placenta. They apply current data mining and analysis methods to identify protein-centred networks that are likely enriched to specific cell types of the placenta. They then translate these findings to humans using statistical comparisons of human single-cell sequencing data of the placenta. Lastly, they use knock-down experiments to validate the conserved functional importance of the hub genes in the mouse protein networks in human cells.

The strengths of this paper are the rigorous data mining methods and the functional translation to humans from mice. There are no critical weaknesses to the article. There is a blend of statistical analysis with anecdotal or hand curation from databases and the literature, but it is unclear if these curated findings are circumstantial or statistically meaningful. In the end, the hypothesis seems to hold in that 4/4 gene knocked down in the human cells gave a migration phenotype.

Comments, questions, critique:

1. Given the translational aims of the paper, more introduction/discussion material on the comparative aspects of mice and humans are needed. Are giant cells and EVT the same? What are the cell equivalents that you are discovering? The Soncin et al. paper is cited, but I think underused. This publication contains time series data on mice and humans and could be used as external validation of clusters, networks, and other analyses. Other publications to consider for context are

a. Cox B, et al. *Mol Syst Biol* 5: 279.

b. Silva JF, Serakides R. 2016. *Cell Adhes Migr* 10: 88-110. (specifically discusses migration difference between the species placentae)

We thank the reviewer for the comment and valuable resources. We agree that more information about the similarities and differences between the migratory cells needs to be provided. We have added the following details in the introduction of the manuscript:

“Although there are certain differences between the mouse and human placenta (Hemberger, Hanna, and Dean 2020; Soncin, Natale, and Parast 2015), they do express common genes during gestation, including common regulators and signaling pathways involved in placental development (Cox et al. 2009; Soncin et al. 2018; Soncin, Natale, and Parast 2015; Watson and Cross 2005). For example, *Ascl2/ASCL2* and *Tfap2c/TFAP2C* are required for the trophoblast (TB) cell lineage in both mouse and human models (Guillemot et al. 1994; Kuckenber, Kubaczka, and Schorle 2012; Varberg et al. 2021). Another example is the HIF signaling pathway, which regulates TB differentiation in both mouse and human placenta (Soncin, Natale, and Parast 2015).”

“Although the structure of the placenta is not identical between mouse and human, certain mouse placental cell types are thought to be equivalent to human placental cell types (Soncin, Natale, and Parast 2015). For example, parietal TGCs and glycogen TBs have been described as equivalent to human extravillous trophoblasts (EVTs) (Soncin, Natale, and Parast 2015). Mouse TGCs are not as invasive as human EVT (Soncin, Natale, and Parast 2015), and they have different levels of polyploidy and copy number variation (Morey et al. 2021); however, both EVT and TGCs are able to degrade extracellular matrix to enable TB migration into the decidua (Silva and Serakides 2016).”

Added to discussion:

“These genes were selected primarily based on the network analyses, but also based on expression data from human cells to account for possible differences between mouse and human placental gene expression.”

Full Revision

As the reviewer suggested, we used the Soncin et al., 2015 data for validation. Only 6,317 of the 11,713 protein-coding genes used for hierarchical clustering were detected in the mouse dataset in Soncin et al., 2015. This issue could be because the Soncin data was generated using microarrays.

Nevertheless, we still compared our e7.5 and e9.5 hierarchical groups with: (1) Soncin et al. gene clusters in mouse that were downregulated over time, had highest expression from e9.5-12.5, or were upregulated over time; and (2) Soncin et al. gene clusters in human that were best correlated with mouse clusters and were either downregulated over time or upregulated over time. We observed a general consensus that our e7.5-hierarchical group had the highest percent of agreement with Soncin et al. gene groups that are downregulated over time, and our e9.5-hierarchical group had the highest percent of agreement with Soncin et al. gene groups that either have highest expression at e9.5-e12.5 or genes that are upregulated over time. This data is added below, described in the results section 1, and included in Supplementary Table S1.

Comparison with Soncin et al. mouse data:

	Having expression > 0 (in Soncin et al.) and being in any hierarchical clusters	E7.5-hierarchical genes (down-regulation trend)	E9.5-hierarchical genes (up-regulation trend)
Cluster 2, 3 and 7 (Soncin et al., downregulation trend)	1009	800 (79.3%)	279 (27.7%)
Cluster 6 (Soncin et al., highest at e9.5 – e12.5)	120	51 (42.5%)	110 (91.7%)
Cluster 1, 4 and 5 (Soncin et al., upregulation trend)	1019	415 (40.7%)	881 (86.5%)

Comparison with Soncin et al. human data:

	Having expression > 0 (in Soncin et al.) and being in any hierarchical clusters	E7.5-hierarchical genes (down-regulation trend)	E9.5-hierarchical genes (up-regulation trend)
HS Cluster 5 (Soncin et al., downregulation trend)	164	92 (56.1%)	52 (31.7%)
HS Cluster 2 and 4 (Soncin et al., upregulation trend)	111	44 (39.6%)	72 (64.9%)

The following statement was added to the result section:

“Second, we compared our hierarchical groups with previously published mouse and human placental microarray time course data from Soncin et al., 2015 (Soncin, Natale, and Parast 2015). Despite the technical differences between the datasets, we observed a consensus that our e7.5 hierarchical cluster had the highest percent of overlap with Soncin et al. gene groups that are downregulated over time, and our e9.5 hierarchical cluster had the highest percent of overlap with Soncin et al. gene groups that either have highest expression at e9.5 - e12.5 or genes that are upregulated over time (Supplementary Table S1).”

2. Clustering represented in Figure 1B, was this a supervised model? Why only three clusters?) Did you specify that there would be three models and force each gene profile into one of the categories? How robust are the fits? A fitted model might be a better approach as you can specify the ideal models (early high, late high and mid-high), then determine each gene profile that fits each model and only assess those genes with a significant fit to the model. Forcing clustering to the three-model fit likely gives many poorly fitting profiles. While in the end, this works out, it may be due to applying other post hoc methods for gene enrichment, where noise distributes randomly.

We carried out unsupervised transcript clustering using hierarchical clustering (agglomerative approach using Euclidean distance and complete linkage). The resulting dendrogram was cut at

the second highest level to obtain three clusters. We have added additional validation with different numbers of clusters ($k = 3, 4$ and 5) and quantification of agreement between different clustering methods to show the robustness of the hierarchical clusters. We acknowledge that hierarchical clustering could be sensitive to noise and could result in poorly fitted transcripts in each group; however, it was a necessary first step for us to identify genes relevant to the distinct placental processes at the three timepoints. Acknowledging this disadvantage, we only focused the analyses on genes that are differentially expressed over time and were present in the timepoint hierarchical groups.

We added the additional analysis as Supplementary Figure S1, and the following statements were added in the results section:

"First, we used three different algorithms, K-means clustering, self-organizing maps, and spectral clustering, to validate the trends of the expression levels in hierarchical groups, as well as the number of transcript groups ($k = 3, 4$ and 5). Only with $k = 3$ did we obtain groups with median expression level trends consistent in all four algorithms (Supplementary Figure S1). Moreover, with $k = 3$, the maximum percent of agreement (see Materials and Methods) between hierarchical clusters and clusters obtained using each of the different algorithms was 70.34-87.26% (Supplementary Figure S1), while the maximum percent of agreement between hierarchical clusters and clusters obtained from other algorithms decreases to between 55.67-65.72% with $k = 4$ and 54.81-59.19% with $k = 5$."

We agree model-based clustering could be an alternative approach and have added it to the discussion section:

"Combining hierarchical clustering with differential expression analysis, we were able to identify gene groups using an unsupervised approach. It has also been shown that for times-series analyses with fewer than eight timepoints, pairwise differential expression analysis combined with additional methods identifies a more robust set of genes (Spies et al. 2019). Alternatively, model-based clustering using RNA-seq profiles (Si et al. 2014) could also be useful for gene group identification. However, it is still important to evaluate the robustness and functional relevance of the fitted models by carrying out additional downstream analyses."

3. Several statements are made about the conservation of importance between mouse and human hub genes. For example, "We predict these highly expressed genes to be generally important for TB function and processes such as cell migration, a term associated with multiple timepoint specific networks (Figure 2A)." While your knock-down assay of migration results

shows these hub genes to be necessary to humans, what do they mean to the mouse? You did not use mouse TSC to assess functional importance concurrently. You note a small number of genes as of known importance, "127 hub genes of which 16 have been annotated as having a role in placental development". Were the others knocked out but lack a developmental phenotype or not assessed? Are these functionally redundant in the mouse or not involved in the same processes between the species?

To assess the possible role of hub genes in mouse development more comprehensively, we extended our search for gene functions on the Mouse Genome Informatics (MGI) database to include not only placenta related GO and MGI phenotype terms (defined as "genes with known roles"), but also embryo related GO and MGI phenotype terms (defined as "genes with possible roles"). We included embryo related terms as "genes with possible roles" because embryonic lethal mouse knockout lines frequently have placentation defects, and because defects in placental development can be associated with the development of other embryonic tissues (Brown and Hay 2016; Perez-Garcia et al. 2018; Woods, Perez-garcia, and Hemberger 2018). This change resulted in an increase in the number of genes with relevant functions in mouse, including several annotated as embryonic lethal or with abnormal embryonic growth (see Supplementary Table S6). With the additional annotations:

- 6 out of 17 hub genes of e7.5 networks have known/possible roles.
- 17 out of 28 hub genes of e8.5 networks have known/possible roles.
- 48 out of 127 hub genes of e9.5 networks have known/possible roles.

We also carried out randomization tests to determine if the number of known/possible genes we identified were significant. Randomization tests were carried out with the following procedure: for each timepoint, from the respective timepoint-specific groups, we sampled 10,000 gene sets of the same number as the hub gene numbers. Then we counted the number of known/possible genes in each random set. A p-value is calculated as the number of times a random gene set has \geq known/possible genes than the observed number, divided by 10,000. We found that the number of genes with known/possible roles at each time point are statistically significant (Supplementary Figure S3). This result indicates that the gene sets we identified are significantly associated with relevant phenotypes in mouse.

The remaining hub genes are unannotated as related to placental or embryonic functions in the MGI database. Based on that, it is difficult to determine if they lack a relevant phenotype, or if there has not been a detailed assessment of the placenta.

Added to section 2 of the result section:

“Briefly, genes annotated under any GO or MGI phenotype terms related to placenta, TB cells, TE and the chorion layer are considered as having a “known” role in the placenta. Genes annotated under terms related to embryo are considered as having a “possible” role in the placenta, because embryonic lethal mouse knockout lines frequently have placentation defects, and because defects in placental development can be associated with the development of other embryonic tissues (Brown and Hay 2016; Perez-Garcia et al. 2018; Woods, Perez-garcia, and Hemberger 2018). Hereafter, such genes are referred to as “known/possible genes”. In the e7.5 networks, there were 17 hub genes in which six genes were known/possible. The number of hub genes that are labelled as known/possible is statistically significant when comparing to random gene sets selected from the e7.5 timepoint-specific group (Supplementary Figure S3). In the e8.5 and e9.5 networks, 17 out of 28 and 48 out of 127 hub genes were known/possible, respectively. Similar to e7.5, the number of hub genes labelled as known/possible in e8.5 networks and e9.5 networks were both statistically significant when comparing to random gene sets selected from the corresponding timepoint-specific groups (Supplementary Figure S3). These results indicate that the gene sets we identified are significantly associated with relevant phenotypes in the mouse.”

For the four genes that we tested in HTR-8/SVneo cells, we also added more information about the current known role of the gene in mouse.

Added to the discussion section:

“We identified hub genes and their immediate neighboring genes which could regulate placental development and confirmed the roles of four novel genes (*Mtdh*, *Siah2*, *Hnrnpk* and *Ncor2*) in regulating cell migration in the HTR-8/SVneo cell line. These genes were selected primarily based on the network analyses, but also based on expression data from human cells to account for possible differences between mouse and human placental gene expression. Previous studies suggested these four candidates are functionally important in mouse. *Mtdh* has been suggested to regulate cell proliferation in mouse fetal development (Jeon et al. 2010). The *Siah* gene family is important for several functions (Qi et al. 2013). Of relevance to the placenta, *Siah2* is an important regulator of HIF1 α during hypoxia both *in vitro* and *in vivo* (Qi et al. 2008). Moreover, while *Siah2* null mice exhibited normal phenotypes, combined knockouts of *Siah2* and *Siah1a* showed enhanced lethality rates, suggesting the two genes have overlapping modulating roles (Frew et al. 2003). *Hnrnpk*^{-/-} mice were embryonic lethal, and *Hnrnpk*^{+/-} mice had dysfunctions in neonatal survival and development (Gallardo et al. 2015). *Ncor2*^{-/-} mice were embryonic lethal before e16.5 due to heart defects (Jepsen et al. 2007). According to the

International Mouse Phenotyping Consortium database (Dickinson et al. 2016), *Ncor2* null mice also showed abnormal placental morphology at e15.5. However, none of these genes have been studied in TB migration function.”

4. In determining conservation between mouse and human networks, were only 1:1 orthologs examined or did you consider more complex 1:many mapping conditions between the two species?

In this work, we used only one-to-one orthology between mouse and human avoid duplication while sampling in the enrichment tests. We added this detail in the method section. However, as found in Cox et al., 2009, genes with one-to-many orthologs could be highly intriguing and should be investigated in future studies.

5. Should the migration assay be normalized to survival/adhesion? If 70,000 cells were seeded but had 50% cell death (or reduced adhesion), then it may appear to be poor migration. Should the migration be evaluated as a ratio of top to bottom cell densities to control for poor adhesion or survival?

We thank the reviewer for bringing up this important point. Unfortunately, with the method we used we cannot quantify the densities on top, because the cells on top need to be scraped off prior to measuring the cells at the bottom (the two densities cannot be measured separately). To help with this concern, in a separate experiment we instead counted cell numbers 48-hours post-transfection for cells treated with target gene siRNA and cells treated with negative control siRNA to determine if apoptosis or changes in proliferation rate could be leading to changes in the observed migration. From this data, we determined that none of the siRNA knockdowns resulted in a significant change of cell counts (p-value > 0.05). We do note that *Siah2* siRNA #1 has some decrease in counts (p-value = 0.081) and *Ncor2* siRNA #1 and #2 have some increase in cell counts (p-value = 0.081 and p-value = 0.077) (Supplementary Figure S7). Additional follow up experiments we have performed with our targets of interest, which are out of the scope of this paper, demonstrate that different pathways and processes could be involved in the resulting decrease in migration we observed (we are following up experimentally in more detail for each gene). Proliferation and other assays could also be used to further examine the increase in *Ncor2* cell counts that were observed. We have added the cell count results and additional text to the discussion.

Added to results, section 4:

“When comparing the number of cells 48 hours post-transfection for cells treated with target gene siRNA to cells treated with negative control siRNA, we determined that none of the target gene siRNA treatments resulted in significant changes in cell counts. We do note that *Siah2* siRNA #1 has some decrease in cell counts (p-value = 0.081), and *Ncor2* siRNA #1 and *Ncor2* siRNA #2 have some increase in cell counts (p-value = 0.081 and p-value = 0.077) compared to negative control treated samples (Supplementary Figure S7). This provides evidence that, in general, the reduction in cell migration capacity was likely not due to the target gene impacting the rate of cell death.”

To the discussion:

“Moreover, we observed that cell counts generally were not decreased upon target gene knockdown compared to negative control knockdown. However, more detailed analysis and process specific assays are needed. For example, future studies assessing each gene’s role in cell adhesion, cell-cell fusion, cell proliferation and cell apoptosis can be done to better understand their roles in placental development.”

Reviewer #1 (Significance (Required)):

This significantly advances previous publications on this topic by functionally testing the discovered genes.

This highlights an excellent data mining strategy for a developmental disease using mice and translating to humans.

The audience is likely developmental biologists and reproductive specialists.

My expertise is bioinformatics and developmental biology.

Reviewer #2 (Evidence, reproducibility and clarity (Required)):

The authors used RNA-seq data from mouse fetal placenta at e7.5, e8.5, and e9.5 to create timepoint-specific gene expression interaction networks to find genes that they predicted would regulate placental development. They confirmed four novel candidate genes and showed that in the transfected human trophoblast HTR-8/SVneo cell line, these four candidates reduced cell migration capacity. Additionally, the authors show that bulk RNA-seq data can be used to infer cell-type composition and when used with single-cell RNA-seq, can be a powerful tool to study the biological processes that involve multiple cell-types.

Overall, the authors are rigorous in their analyses, their conclusions appear sound, and the work could be an asset to the broader placental biology field. However, although the authors present an approach that future studies might find useful to replicate and their work has produced numerous novel transcripts/genes that warrant further investigation, the approach is not entirely novel, and could be expanded/improved (as suggested by the authors in the discussion),

particularly with regard to validation of the genes/networks identified. Major and minor comments are listed below.

Major comments:

1) The authors used clustering and differential expression analysis to define sets of timepoint-specific genes. However, it was not clear to me the benefits of this approach. Why would using this approach be better than differential expression analysis alone such as in a typical ANOVA?

We have added more discussion on this matter to explain our approach. We believe using hierarchical clustering and pairwise differential expression analysis can help identify gene lists with higher confidence. These are the new details we added to the discussion section:

“Combining hierarchical clustering with differential expression analysis, we were able to identify gene groups using an unsupervised approach. It has also been shown that for times-series analyses with fewer than eight timepoints, pairwise differential expression analysis combined with additional methods identifies a more robust set of genes (Spies et al. 2019). Alternatively, model-based clustering using RNA-seq profiles (Si et al. 2014) could also be useful for gene group identification. However, it is still important to evaluate the robustness and functional relevance of the fitted models by carrying out additional downstream analyses.”

2) Related to number 1 above, although the authors are interested in timepoint-specific transcripts, the author's methods would filter out possibly interesting transcripts that turn on and off during development. The authors might want to check to see if there are transcripts that are up in e7.5 and then down in e8.5 but then up again in e9.5. Also, the author's methods seem to include transcripts that are not exclusive to one timepoint (i.e. are up in e7.5 and e8.5 but not e9.5). It might be interesting to differentiate transcripts that are exclusive to one timepoint from those that are in more than one timepoint.

We thank the reviewer for their valuable comment. We agree genes that turn on and off during the time course could be very interesting. In performing this analysis, we found that the number of such genes is rather small (38 genes that are up-regulated at e7.5 compared to e8.5 and up-regulated at e9.5 compared to e8.5). These genes were not enriched for processes that we observed with timepoint-specific gene groups, such as “trophoblast giant cell differentiation” (e7.5-specific genes), “labyrinthine layer development” (e8.5- and e9.5-specific genes), “blood vessel development” (e7.5- and e9.5-specific genes) and “response to nutrient” (e9.5-specific genes) (Supplementary Table S3). They are generally enriched for processes related to cytokine production and regulation of secretion.

We also agree that it is interesting to differentiate transcripts that are exclusive to one time point from those that are in more than one time point. In the revised manuscript, we added additional analysis for genes that belong to multiple timepoint groups due to different transcripts of the same gene being annotated as timepoint-specific, and genes unique to each timepoint (Added to results section 1):

“It is possible that timepoint-specific groups share genes that have timepoint-specific transcripts. Indeed, we identified 37 genes shared between e7.5 and e8.5, 5 genes shared between e7.5 and e9.5, and 109 genes shared between e8.5 and e9.5 (Supplementary Table S3). We found that genes only present at one timepoint (timepoint-unique genes) were generally enriched for similar terms as the full group of timepoint-specific genes (Supplementary Table S3). However, terms related to the development of labyrinth layer like “labyrinthine layer morphogenesis” and “labyrinthine layer blood vessel development” were only enriched when using all e8.5-specific genes but not when using e8.5 timepoint-unique genes. Moreover, we found that, unlike genes shared between e9.5 and e7.5, genes shared between e9.5 and e8.5 were enriched for processes such as “blood vessel development” and “insulin receptor signaling pathway”. This observation may indicate that different transcripts of the same genes could be expressed at different timepoints for the continuation of certain biological processes.”

3) In the network analysis it would be interesting and helpful to the reader to highlight, if any, nodes or terms that were found to be significant (i.e. hubs or genes that have a high centrality metric etc.) in both the STRING and GENIE3 networks or overlap the networks created by the two different algorithms to compare them. This might help readers better rank genes when using these data to decide what genes are most important at each timepoint.

We observed only one hub gene shared among networks inferred by the two methods (*Vegfa* in the e9.5 networks). However, hub genes of networks inferred by one method could be nodes in networks inferred by the other method. Hence, we have added lists of such genes in section 2. Interestingly, many of these genes have known roles in placental development. In terms of biological functions shared between the networks at the same timepoints, there were multiple interesting processes such as “positive regulation of cell migration”, “epithelium migration” and “vasculature development”, which we highlighted in Figure 2A.

In the revised manuscript, we have added the following details in different paragraphs of section 2 of the results:

“Although the networks inferred by the two methods did not share any hub genes, hub genes identified with one method could be members of the other method’s networks. These hub genes are *Mmp9* (e7.5_1_STRING), *Frk*, *Hmox1*, and *Nr2f2* (e7.5_2_GENIE3) (Table 1). This observation strengthens the potential roles of *Frk* gene in placental development.”

“Hub genes identified with one method and present in the other method’s networks are *Hsp90aa1*, *Akt1*, and *Mapk14* (e8.5_1_STRING), *Dvl3* and *Msx2* (e8.5_2_GENIE3) (Table 1).”

“Hub genes identified with one method and present in the other method’s networks include important genes such as *Rb1* (Sun et al. 2006), *Yap1* (Meinhardt et al. 2020) (e9.5_1_GENIE3) and *Vegfa* (e9.5_2_STRING) (Table 1). Notably, *Vegfa* is the only hub gene identified with both of the network inference methods.”

4) The author's conclusion that network analysis can be used to identify genes more likely associated with specific placental cell types is very likely true, but I think that the conclusion would be more impactful if the authors reported how the method compares to simply taking a list of differentially expressed genes and looking for cell type enrichments using their favorite enrichment software. For example, if a gene is highly connected in a particular network that has been identified as SCT-specific, but that gene isn't considered an SCT "marker" by the placental biology research community, it would be interesting to highlight that it is prevalent in a previously published scRNA-seq dataset or a dataset that has isolated that particular cell type to show the advantages of using networks to find placental cell type specific genes.

We completely agree with the reviewer’s point and have now added a randomization analysis to compare the enrichment using PlacentaCellEnrich (PCE) with genes in networks and random genes (Supplementary Figure S6). We randomly sampled 10,000 gene sets with the same sizes as the subnetworks from their corresponding hierarchical groups and carried out PCE analysis. These tests showed that the enrichments of cell type-specific genes were only significant with the subnetwork genes but not the random genes. The randomization tests added a valuable highlight that the network genes are highly relevant to cell type-specific genes in the human placenta, and therefore provided more confidence in the gene lists obtained from the network analyses.

We also further checked the expression of the hub genes in other independent data in order to identify hub genes that are potentially cell type specific markers. For example, we observed that *Dvl3* (e8.5_2_GENIE3) and *Olr1* (e9.5_3_STRING) have been shown to be differentially

expressed in SCT compared to other TB subtypes (human trophoblast stem cells, EVT (Sheridan et al. 2021) or endovascular TB (Gormley et al. 2021)).

We added the following detail in the results, section 3:

“Importantly, randomization tests showed that the enrichment of cell type-specific genes were only significant in these subnetworks but not in random gene sets selected from corresponding timepoint hierarchical groups (Supplementary Figure S6), which highlights the biological relevance of the gene network modules.”

Added to the discussion section:

“Moreover, hub genes could be used to identify potential novel markers for the cell types corresponding to their subnetworks. For example, hub genes of subnetworks enriched for SCT-specific genes such as *Dvl3* (e8.5_2_GENIE3) and *Olr1* (e9.5_3_STRING) are not established SCT marker genes, but are in fact differentially expressed in SCT compared to human trophoblast stem cells, EVT (Sheridan et al. 2021) or endovascular TB (Gormley et al. 2021). In general, combining network analysis with existing gene expression data from single cell or pure cell populations will allow identification of novel cell-specific marker genes to help future studies focused on different TB populations.”

5) While the selection of genes for validation was limited by the model system available for testing, the authors should recognize that the genes/networks identified here should first and foremost be validated in a mouse model (by knockdown/overexpression studies using mouse trophoblast stem cells or by evaluation of placenta/embryo in a KO/transgenic mouse model). Whether or not the data are relevant to human placentation is (at least initially) irrelevant. While we recognize that these are difficult studies requiring significant time and resources, as is, the data and results will have significantly less impact than if even a limited amount of such validation could be performed.

We thank the reviewer for this valuable comment. Based on this comment and the suggestions from reviewer #1, we have added the following points to the manuscript to discuss the relevance of the genes in the mouse models, and further explain our gene choices:

To assess the possible role of hub genes in mouse development more comprehensively, we extended our search for gene functions on the Mouse Genome Informatics (MGI) database to include not only placenta related GO and MGI phenotype terms (defined as “genes with known roles”), but also embryo related GO and MGI phenotype terms (defined as “genes with possible roles”). We included embryo related terms as “genes with possible roles” because embryonic

lethal mouse knockout lines frequently have placentation defects, and because defects in placental development can be associated with the development of other embryonic tissues (Brown and Hay 2016; Perez-Garcia et al. 2018; Woods, Perez-garcia, and Hemberger 2018). This change resulted in an increase in the number of genes with relevant functions in mouse, including several annotated as embryonic lethal or with abnormal embryonic growth (see Supplementary Table S6). With the additional annotations:

- 6 out of 17 hub genes of e7.5 networks have known/possible roles.
- 17 out of 28 hub genes of e8.5 networks have known/possible roles.
- 48 out of 127 hub genes of e9.5 networks have known/possible roles.

We also carried out randomization tests to determine if the number of known/possible genes we identified were significant. Randomization tests were carried out with the following procedure: for each timepoint, from the respective timepoint-specific groups, we sampled 10,000 gene sets of the same number as the hub gene numbers. Then we counted the number of known/possible genes in each random set. A p-value is calculated as the number of times a random gene set has \geq known/possible genes than the observed number, divided by 10,000. We found that the number of genes with known/possible roles at each time point are statistically significant (Supplementary Figure S3). This result indicates that the gene sets we identified are significantly associated with relevant phenotypes in mouse.

The remaining hub genes are unannotated as related to placental or embryonic functions in the MGI database. Based on that, it is difficult to determine if they lack a relevant phenotype, or if there has not been a detailed assessment of the placenta.

Added to section 2 of the result section:

“Briefly, genes annotated under any GO or MGI phenotype terms related to placenta, TB cells, TE and the chorion layer are considered as having a “known” role in the placenta. Genes annotated under terms related to embryo are considered as having a “possible” role in the placenta, because embryonic lethal mouse knockout lines frequently have placentation defects, and because defects in placental development can be associated with the development of other embryonic tissues (Brown and Hay 2016; Perez-Garcia et al. 2018; Woods, Perez-garcia, and Hemberger 2018). Hereafter, such genes are referred to as “known/possible genes”. In the e7.5 networks, there were 17 hub genes in which six genes were known/possible. The number of hub genes that are labelled as known/possible is statistically significant when comparing to random gene sets selected from the e7.5 timepoint-specific group (Supplementary Figure S3). In the e8.5 and e9.5 networks, 17 out of 28 and 48 out of 127 hub genes were known/possible,

respectively. Similar to e7.5, the number of hub genes labelled as known/possible in e8.5 networks and e9.5 networks were both statistically significant when comparing to random gene sets selected from the corresponding timepoint-specific groups (Supplementary Figure S3). These results indicate that the gene sets we identified are significantly associated with relevant phenotypes in the mouse.”

For the four genes that we tested in HTR-8/SVneo cells, we also added more information about the current known role of the gene in mouse.

Added to the discussion section:

“We identified hub genes and their immediate neighboring genes which could regulate placental development and confirmed the roles of four novel genes (*Mtdh*, *Siah2*, *Hnrnpk* and *Ncor2*) in regulating cell migration in the HTR-8/SVneo cell line. These genes were selected primarily based on the network analyses, but also based on expression data from human cells to account for possible differences between mouse and human placental gene expression. Previous studies suggested these four candidates are functionally important in mouse. *Mtdh* has been suggested to regulate cell proliferation in mouse fetal development (Jeon et al. 2010). The *Siah* gene family is important for several functions (Qi et al. 2013). Of relevance to the placenta, *Siah2* is an important regulator of HIF1 α during hypoxia both *in vitro* and *in vivo* (Qi et al. 2008). Moreover, while *Siah2* null mice exhibited normal phenotypes, combined knockouts of *Siah2* and *Siah1a* showed enhanced lethality rates, suggesting the two genes have overlapping modulating roles (Frew et al. 2003). *Hnrnpk*^{-/-} mice were embryonic lethal, and *Hnrnpk*^{+/-} mice had dysfunctions in neonatal survival and development (Gallardo et al. 2015). *Ncor2*^{-/-} mice were embryonic lethal before e16.5 due to heart defects (Jepsen et al. 2007). According to the International Mouse Phenotyping Consortium database (Dickinson et al. 2016), *Ncor2* null mice also showed abnormal placental morphology at e15.5. However, none of these genes have been studied in the context of TB migration.”

Minor comments:

1) In the GO analysis, why not use a combination of hypergeometric and binomial distribution for enrichment decisions?

We used hypergeometric tests as in the default setting of ClusterProfiler. GO enrichment with hypergeometric test for differentially expressed genes was also suggested in Rivals et al., 2007 (Rivals et al. 2007). Combination of hypergeometric and binomial tests will be of great use when

carrying out enrichment for cis-regulatory domains where there is a higher chance of sampling a gene randomly (McLean et al. 2010).

We have added this detail in the method section to make the analysis clearer.

2) In Figure 2B, are there any genes that are both hub nodes (diamonds) and annotated as having placental functions (squares)? If so, it might be good to show that in some way.

We agree this is necessary and have altered the presentation in Figure 2. In the revised manuscript, we have added an additional list of hub genes as genes with possible roles. The figure now shows hub genes with known placental functions (diamonds), hub genes with possible functions (hexagons) and hub genes without related annotation (rounded squares). Non-hub genes are now not shown to avoid crowdedness.

3) It might improve the deconvolution analysis to employ more than one method and recent reports have shown that the cell-type signature data is the most important parameter with the main factors influencing performance being biological (such as where the sample was taken) rather than technical (<https://doi.org/10.1038/s41467-022-28655-4>).

We agree the conclusion would have been further confirmed if we could employ another deconvolution method. Upon literature search, we found another tool, CAM (N. Wang et al. 2016), that had similar approaches to LinSeed which aims to infer cell proportions without reference. However, the tool has been taken down from Bioconductor and is not currently maintained. As a result, to the best of our knowledge, LinSeed is the only deconvolution tool that is completely reference-free.

We also tried carrying out the deconvolution analysis with another method, DSA (Zhong et al. 2013), with a limited number of marker genes obtained through literature review. However, when the marker genes are highly correlated in multiple cell types, the models failed to infer meaningful proportions.

We acknowledge that we need additional single cell RNA-seq data or marker genes obtained from pure cell populations to make more concrete conclusions for the deconvolution analysis. We hope with future studies, there will be more evidence supporting our observations.

We have added this acknowledgement in the results section:

“The identification of these cell groups could have resulted from noise introduced by both biological and technical variation, which is challenging to overcome when using a small sample size or analyzing without prior knowledge in the deconvolution analysis.”

Added to the discussion section:

“Nevertheless, we acknowledge that our deconvolution analysis and cell type annotations were limited due to the absence of matching scRNA-seq data, data from pure cell populations, or extensive cell marker lists. As these types of information are available, deconvolution analysis can be used to identify species-specific cell types or correcting for confounding effects prior to DEA (Sutton et al. 2022).”

4) The above report also shows that there are ways to correct for cell-type composition differences in DEA which might be interesting to look when using bulk data from different timepoints in future studies when focusing on different biological processes and not timepoint-specific transcripts.

We agree correcting for cell proportion prior to differential expression analysis will be interesting for future studies. When single cell RNA-seq data or more extensive marker gene lists are available, deconvolution analysis will be of great use for this purpose.

We have added this in the discussion section (also mentioned in point #3):

“Nevertheless, we acknowledge that our deconvolution analysis and cell type annotations were limited due to the absence of matching scRNA-seq data, data from pure cells, or extensive cell marker lists. As these types of information become more available, deconvolution analysis can be used to identify species-specific cell types or correcting for confounding effects prior to DEA (Sutton et al. 2022).”

5) Could the authors speculate as to possible reason(s) that an siRNA knockdown would give variable results functionally, while the actual gene expression appears to be consistently and sufficiently downregulated? Did the authors evaluate protein levels following siRNA knockdown?

Following the reviewer’s comment, we have evaluated protein levels for each target gene and each siRNA. For the genes that gave variable results between siRNAs (MTDH and NCOR2), we did not observe a change in their ability to reduce protein levels (Supplementary Figure S7). It is therefore possible that there are off-target effects for one of the siRNAs. We considered this possibility in designing the project, which is why we tested two siRNAs per target gene.

Although siRNA off-target effects may be present, visual inspection of the migration experiments indicate that transfection with each of the siRNAs reduces migration capacity. We have added the possibility of off-target effects in the discussion section:

“We observed that while all siRNAs were able to decrease cell migration capacity, there was variability in the amount of decrease, even when comparing two siRNAs targeting the same gene. This observation did not seem to be associated with differences in transcript or protein knockdown levels and could be due to different off-target effects for different siRNAs.”

6) As mentioned in the discussion, finding genes that have timepoint dependent isoforms would be an interesting and novel addition to the manuscript.

Protein isoforms would be interesting to study. Here we focused on different mRNA transcripts. We carried out additional GO analysis on the genes unique to each timepoint and genes shared among timepoints. This was also done in response to major comment 2:

In the revised manuscript, we added additional analysis for genes that belong to multiple timepoint groups due to different transcripts of the same gene being annotated as timepoint-specific, and genes unique to each timepoint (Added to results section 1):

“It is possible that timepoint-specific groups share genes that have timepoint-specific transcripts. Indeed, we identified 37 genes shared between e7.5 and e8.5, 5 genes shared between e7.5 and e9.5, and 109 genes shared between e8.5 and e9.5 (Supplementary Table S3). We found that genes only present at one timepoint (timepoint-unique genes) were generally enriched for similar terms as the full group of timepoint-specific genes (Supplementary Table S3). However, terms related to the development of labyrinth layer like “labyrinthine layer morphogenesis” and “labyrinthine layer blood vessel development” were only enriched when using all e8.5-specific genes but not when using e8.5 timepoint-unique genes. Moreover, we found that, unlike genes shared between e9.5 and e7.5, genes shared between e9.5 and e8.5 were enriched for processes such as “blood vessel development” and “insulin receptor signaling pathway”. This observation may indicate that different transcripts of the same genes could be expressed at different timepoints for the continuation of certain biological processes.”

7) Although outside the scope of this manuscript, it might be interesting to look at the effects of knocking down network genes on the networks themselves and in combination with a phenotypic readout such as a migration assay. With numerous knockouts and migration assay readouts, one could possibly find a better method to rank the genes within the networks.

We agree with this comment. Upon literature search, we realized this approach has been used in previous studies on other biological contexts such as virus entry (A. Wang et al. 2010; A. Wang, Ren, and Li 2011) and cancer cell growth (Paul et al. 2021). Although these studies used

different network inference strategies from ours, their *in silico* gene knockouts proved to be effective for the candidate selection. However, the knockout process (both computationally and experimentally) may not be trivial; therefore, we agree the approach will be useful for future studies.

CROSS-CONSULTATION COMMENTS

I mostly agree with the other two reviewers.

It is not clear to me that additional KD experiments (i.e. ones that might affect fusion, proliferation, apoptosis), as proposed by Reviewer #3, would be that much more informative. There are many differences between mouse and human placentation, and these model systems (HTR8 and BeWo) are not truly representative of either. The additional data mining/computational work would be more useful and enhance data interpretation.

Reviewer #2 (Significance (Required)):

The authors use RNA-seq of mouse placenta at e7.5, e8.5, and e9.5 to show that timepoint-specific expression patterns are highly correlated with certain biological processes and point to the existence of certain cell types in the sample. While focused on early post-implantation mouse placental development, the author's methods could be transferrable to other timepoints, species, and organs. Furthermore, with their method they uncover what appears to be several novel, early placental, developmentally important genes and their results might be of interest to those in the field studying placental development.

Reviewer #3:

Summary:

This paper is an analysis of RNA-seq data from the mouse human placenta at embryonic day from 7.5 to 9.5 days. Bioinformatics was used to pinpoint genes networks, and tentatively connect with human cell populations. Wet experiments were performed on the HTR8/SV neo trophoblast cell model.

The introduction clearly posits the reasons why mouse models were chosen, and presents some examples of genes that are conserved between human and mouse placentas, before presenting the major steps of mouse placental development at the crucial periods analyzed.

The results are divided into four parts:

1. Identification of genes that are specific of fetal tissues at the three days studied
2. A network analysis of the genes using classical bioinformatics tools (String, Genie3) to identify gene modules
3. A connection with the human placenta at the level of cell-specific expression profile is then analyzed
4. A *in vitro* validation on a trophoblast cell model using siRNA to Knockdown genes identified in the *in silico* part of the paper.

Three clustering methods were used to classify the genes according to their profile (at which time point they have the highest level). The function associated are dispatched into three logical

Full Revision

physiological events (7.5: proliferation and ectoplacental cone development, 8.5 attachment of the placenta -chorioallantoidian at this stage- , and 9.5: syncytiotrophoblast constitution and labyrinth development, structures essential for growth and exchange).

Mostly minor comments:

Quality of the transcriptomics data: 6 replicates per condition (some being pools at E7.5 and 8.5) is a lot, and I congratulate the authors to have make such effort. This says a lot about the technical quality of their results. Nevertheless, there is no comment on the exclusion of two samples in the further analysis based upon the PCA. Could the authors comment upon the reasons why these two samples behave so differently from the others?

We thank the reviewer for the comment. We reviewed the RNA concentration and quality prior to sequencing, and did not observe that the outliers were of lower quality. After sequencing, quality control metrics (obtained with FastQC), also did not indicate that the two outliers were of poor quality. Based on the PCA, it is also unlikely that two samples were swapped. One possibility is that the tissues obtained for these samples were diseased in some way. However, this is difficult to confirm, so we did not want to speculate about this in the manuscript. We did exclude the two samples to ensure the accuracy of our downstream analyses.

Rq: at this stage some statistics of the degree of enrichment in keyword should be provided (such as Enrichment Scores, normalized or not, and False Discovery Rates, to be able to evaluate the actual robustness of the genes network identified. In addition, it seems that the authors supervised the 'keywords' and 'ontologies' toward placental function. A more agnostic approach could be very relevant, such as identifying the ontologies associated to for instance the set of genes that are highest at 8.5 days, by comparing them with preliminary datasets accessible via the GSEA platform of the BROAD institute or similar sites such as Webgestalt.

This does not mean that the placental-targeted approach is not useful, but to have a more global overview is in my opinion indispensable.

We agree and this is a good point. We have now added a stringent approach to determine if the placenta-targeted terms are truly relevant to the gene networks. We performed randomization tests using random gene sets sampled from hierarchical groups of the same time point. These tests showed that the selected terms are significant in the networks when compared to gene groups of the same size from the timepoint specific hierarchical groups (Supplementary Figure S3). Moreover, we have added the specific $-\log_{10}(q\text{-value})$ of some highlighted enriched terms in the main text, so together with Figure 2A, the degree of enrichment of these terms can be shown in a clearer way.

We have added this detail in the result section:

“Compared to e8.5 and e9.5 networks, e7.5 networks had a higher rank or fold change and were significantly enriched for the GO terms “inflammatory response” (e7.5_1_STRING: $-\log_{10}(\text{q-value}) = 22.82$ and e7.5_2_GENIE3: $-\log_{10}(\text{q-value}) = 3.95$) and “female pregnancy” (e7.5_2_GENIE3: $-\log_{10}(\text{q-value}) = 4.1$) (Figure 2A, Supplementary Table S5). The term “morphogenesis of a branching structure”, which can be expected following chorioallantoic attachment around e8.5, was not enriched at e7.5, but was enriched in multiple e8.5 and e9.5 networks (e8.5_1_STRING: $-\log_{10}(\text{q-value}) = 1.73$, e8.5_2_GENIE3: $-\log_{10}(\text{q-value}) = 1.72$, e9.5_1_STRING: $-\log_{10}(\text{q-value}) = 4.01$, e9.5_1_GENIE3: $-\log_{10}(\text{q-value}) = 1.54$, e9.5_2_STRING: $-\log_{10}(\text{q-value}) = 14.33$, and e9.5_2_GENIE3: $-\log_{10}(\text{q-value}) = 2.2$). After chorioallantoic attachment finishes, nutrient transport is being established. Accordingly, we observed the following enrichments: “endothelial cell proliferation” (highest ranked in e9.5_2_STRING: $-\log_{10}(\text{q-value}) = 15.91$), “lipid biosynthetic process” (only significant after e7.5, highest ranked in e9.5_3_STRING: $-\log_{10}(\text{q-value}) = 17.63$), “cholesterol metabolic process” (only significant after e7.5, highest ranked in e9.5_2_GENIE3: $-\log_{10}(\text{q-value}) = 2.76$ and e9.5_3_STRING: $-\log_{10}(\text{q-value}) = 7.79$), and “response to insulin” (only significant after e7.5, highest ranked in e9.5_1_GENIE3: $-\log_{10}(\text{q-value}) = 1.67$).”

“Using randomization tests, we observed the majority of these GO terms (10 out of 11 terms) were significantly enriched when using the network genes but not random gene sets (significance level of 0.05; the term “vasculature development” having p-value = 0.0549 and 0.0575 in with subnetwork e9.5_1_GENIE3 and e9.5_3_GENIE3, respectively) (see Materials and Methods, Supplementary Figure S3). This analysis demonstrates that the network genes were highly relevant to the biological functions of interest. Moreover, the observed GO terms strongly aligned with the processes enriched when using the full lists of timepoint-specific genes (Supplementary Table S3), indicating the representative characteristics of the network genes. While the current analysis focuses on the biological processes related to placental development, there are other terms significantly enriched, which can be found in Supplementary Table S5.”

This is partially done in the part 2 of the results, but it would be relevant to do it on the group of highly expressed genes and not only on the clusters found by the algorithm of sting and genie3. We have added GO analysis for timepoint-specific genes and also observed highly relevant processes being enriched (Supplementary Table S3). This additional analysis has also helped strengthen the relevance of the network genes, as the observed terms with network genes aligned well with the terms enriched with the full lists of genes.

Rq: in the second part of the results, everything is descriptive but no hierarchy is given to facilitate the understanding and to try to generate a few 'take-home messages' for the reader.

We agree with the comment and have adjusted the writing accordingly. We have added the following statements in section 2 of the result section:

“In summary, we identified 18 subnetworks across three timepoints for downstream analyses, some of which were enriched, according to GO analysis and randomization tests, for specific terms relating to placental development (Figure 2A).”

“These results indicate that the gene sets we identified are functionally relevant in the mouse models.”

“In summary, we have identified hub genes in networks at each timepoint. Analyzing the annotations of hub genes using the MGI database demonstrated that the hub genes are biologically relevant to mouse development and will be strong candidates for future investigation.”

The network analysis is well presented in Figure 2. I wonder whether the author could add systematically besides the three examples that are given the network analysis for the other enrichment network that are described (the four at e7.5, the 6 at e8.5 and the 8 at e9.5).

We have added the additional figures in Supplementary Figure S3.

The deconvolution of the 3rd part of the results to try to connect the mouse results to the human cell situation is interesting. I suspect that given the terms of the mouse placentas used, it would be relevant to focus on 1st trimester human placental cells.

The reference dataset we used in the PlacentaCellEnrich analysis was from human 1st trimester placenta samples. For the Placenta Ontology analysis, we were limited to the provided database from (Naismith and Cox 2021); however, it will be interesting to revisit the analysis when the database is extended.

We have specified that the reference data in PlacentaCellEnrich analysis was from human 1st trimester placenta in the methods section:

“For PlacentaCellEnrich, cell-type specific groups were based on the single-cell transcriptome data of first trimester human maternal-fetal interface from Vento-Tormo et al.”

As previously mentioned, this is a highly descriptive paragraph, and two or three sentences at the end of each paragraph of the results would be in my opinion indispensable to present the

most important observations of the results in an intelligible way. Overall, the data presented by the authors, are not obviously 'raw data', but an effort of interpretation should be done by the authors to underline the importance of their results, and to stress among these results which are the most important, and which are the most relevant for placental development and human health.

We agree with the comment and have adjusted the writing accordingly. We have added this summary paragraph at the end of section 3 of the result section:

“In summary, we have demonstrated that the identification of timepoint-specific gene groups and densely connected network modules can be used to infer the cellular composition of bulk RNA-seq samples. We used independent human datasets from different sources to annotate the cell types in each timepoint's samples. As a result, from the bulk RNA-seq data we were able to observe that at e7.5 and e8.5, there was a high proportion of different TB populations, whereas at e9.5, the placental tissues consisted of multiple cell types such as TB, endothelial and fibroblast cells.”

In the last part, which is very important in this type of paper, four genes were selected. A choice of highly expressed genes was made (which can in fact be discussed, some transcriptional factors may have a crucial importance with relatively low levels of expression). The efficiency of the siRNA was overall excellent. The authors showed that each of these siRNA is efficient to inhibit cell migration in the HTR8/SVneo model.

The migration assays are quantified, but there is an inherent limit of the cell model: the authors analyzed only cell migration, but not other very important parameters. One of them is trophoblast fusion, an issue that can be studied in another trophoblast cell model, the BeWo cells, which are induced to fuse under forskolin. It would be highly relevant to test the siRNA identified in this respect, since fusion is a very conspicuous feature of trophoblast cells in mice as well as in humans. Other relevant endpoints such as proliferation markers, apoptosis markers, oxidative stress markers could be studied in the KD cell models. Alternatively, it would have been interesting to evaluate the overall effect of the siRNA by transcriptomics and check whether the modified gene expression leads to specific profiles characteristic of a certain moment of placental development in mice, or proportion of various cells in the human placentas. Without asking for further experiments the authors should mention these limits in their discussion.

We completely agree with this comment and are investigating each of our candidate genes in more detail in ongoing studies. As we have already learned that each gene is involved in

different processes and pathways, we feel that these studies are out of the scope of the current paper. However, we have added this point to our discussion section:

“However, more detailed analysis and process specific assays are needed. For example, future studies assessing each gene’s role in cell adhesion, cell-cell fusion, cell proliferation and cell apoptosis can be done to better understand their roles in placental development.”

In sum, I feel that this paper provides an excellent dataset, but that the authors should make an additional effort of redaction to extract the most important conclusions of their paper. This would increase its impact for a wider public.

Thank you. We have attempted to do so in the revised version.

Reviewer #3 (Significance (Required)):

The context is well introduced, but explanatory and synthesis sentences are missing at the end of each paragraph. I am relatively competent in bioinformatics methods, including deconvolution, and rather expert in cell biology. Therefore I feel comfortable to evaluate this paper.

References:

- Brown, Laura D., and William W. Hay. 2016. “Impact of Placental Insufficiency on Fetal Skeletal Muscle Growth.” *Molecular and cellular endocrinology* 435: 69. /pmc/articles/PMC5014698/ (August 24, 2022).
- Cox, Brian et al. 2009. “Comparative Systems Biology of Human and Mouse as a Tool to Guide the Modeling of Human Placental Pathology.” *Molecular Systems Biology* 5: 279. /pmc/articles/PMC2710868/ (July 20, 2022).
- Dickinson, Mary E. et al. 2016. “High-Throughput Discovery of Novel Developmental Phenotypes.” *Nature* 2016 537:7621 537(7621): 508–14. <https://www.nature.com/articles/nature19356> (July 20, 2022).
- Frew, Ian J. et al. 2003. “Generation and Analysis of Siah2 Mutant Mice.” *Molecular and Cellular Biology* 23(24): 9150. /pmc/articles/PMC309644/ (July 27, 2022).
- Gallardo, Miguel et al. 2015. “HnRNP K Is a Haploinsufficient Tumor Suppressor That Regulates Proliferation and Differentiation Programs in Hematologic Malignancies.” *Cancer Cell* 28(4): 486–99. <http://www.cell.com/article/S1535610815003050/fulltext> (August 24, 2022).
- Gormley, Matthew et al. 2021. “RNA Profiling of Laser Microdissected Human Trophoblast

- Subtypes at Mid-Gestation Reveals a Role for Cannabinoid Signaling in Invasion.” *Development (Cambridge, England)* 148(20). <https://pubmed.ncbi.nlm.nih.gov/34557907/> (August 15, 2022).
- Guillemot, François et al. 1994. “Essential Role of Mash-2 in Extraembryonic Development.” *Nature* 371(6495): 333–36. <https://www.nature.com/articles/371333a0> (December 21, 2021).
- Hemberger, Myriam, Courtney W. Hanna, and Wendy Dean. 2020. “Mechanisms of Early Placental Development in Mouse and Humans.” *Nature Reviews Genetics* 21(1): 27–43. <http://dx.doi.org/10.1038/s41576-019-0169-4>.
- Jeon, Hyun Yong et al. 2010. “Expression Patterns of Astrocyte Elevated Gene-1 (AEG-1) during Development of the Mouse Embryo.” *Gene expression patterns : GEP* 10(7–8): 361. </pmc/articles/PMC3165053/> (July 27, 2022).
- Jepsen, Kristen et al. 2007. “SMRT-Mediated Repression of an H3K27 Demethylase in Progression from Neural Stem Cell to Neuron.” *Nature* 450(7168): 415–19. <https://www.nature.com/articles/nature06270> (July 27, 2022).
- Kuckenber, Peter, Caroline Kubaczka, and Hubert Schorle. 2012. “The Role of Transcription Factor Tcfap2c/TFAP2C in Trophectoderm Development.” *Reproductive BioMedicine Online* 25(1): 12–20. <http://www.rbmojournal.com/article/S1472648312001010/fulltext> (December 21, 2021).
- McLean, Cory Y. et al. 2010. “GREAT Improves Functional Interpretation of Cis-Regulatory Regions.” *Nature Biotechnology* 28(5): 495–501. <http://dx.doi.org/10.1038/nbt.1630>.
- Meinhardt, Gudrun et al. 2020. “Pivotal Role of the Transcriptional Co-Activator YAP in Trophoblast Stemness of the Developing Human Placenta.” *Proceedings of the National Academy of Sciences of the United States of America* 117(24): 13562–70. <https://www.ncbi.nlm.nih.gov/geo/> (April 8, 2022).
- Morey, Robert et al. 2021. “Transcriptomic Drivers of Differentiation, Maturation, and Polyploidy in Human Extravillous Trophoblast.” *Frontiers in Cell and Developmental Biology* 9: 2269.
- Naismith, Kendra, and Brian Cox. 2021. “Human Placental Gene Sets Improve Analysis of Placental Pathologies and Link Trophoblast and Cancer Invasion Genes.” *Placenta* 112: 9–15.
- Paul, Abhijit et al. 2021. “Exploring Gene Knockout Strategies to Identify Potential Drug Targets Using Genome-Scale Metabolic Models.” *Scientific Reports* 2021 11:1 11(1): 1–13. <https://www.nature.com/articles/s41598-020-80561-1> (July 27, 2022).

- Perez-Garcia, Vicente et al. 2018. "Placentation Defects Are Highly Prevalent in Embryonic Lethal Mouse Mutants." *Nature* 555(7697): 463. /pmc/articles/PMC5866719/ (August 11, 2022).
- Qi, Jianfei et al. 2008. "The Ubiquitin Ligase Siah2 Regulates Tumorigenesis and Metastasis by HIF-Dependent and -Independent Pathways." *Proceedings of the National Academy of Sciences of the United States of America* 105(43): 16713. /pmc/articles/PMC2575485/ (September 20, 2021).
- Qi, Jianfei, Hyungsoo Kim, Marzia Scortegagna, and Ze'ev A. Ronai. 2013. "Regulators and Effectors of Siah Ubiquitin Ligases." *Cell biochemistry and biophysics* 67(1): 15. /pmc/articles/PMC3758783/ (July 27, 2022).
- Rivals, Isabelle, Lé On Personnaz, Lieng Taing, and Marie-Claude Potier. 2007. "Databases and Ontologies Enrichment or Depletion of a GO Category within a Class of Genes: Which Test?" *Bioinformatics* 23(4): 401–7.
- Sheridan, Megan A. et al. 2021. "Characterization of Primary Models of Human Trophoblast." *Development (Cambridge, England)* 148(21). /pmc/articles/PMC8602945/ (August 15, 2022).
- Si, Yaqing, Peng Liu, Pinghua Li, and Thomas P. Brutnell. 2014. "Model-Based Clustering for RNA-Seq Data." *Bioinformatics* 30(2): 197–205. <https://academic.oup.com/bioinformatics/article/30/2/197/217752> (July 18, 2022).
- Silva, Juneo F., and Rogéria Serakides. 2016. "Intrauterine Trophoblast Migration: A Comparative View of Humans and Rodents." *Cell Adhesion and Migration* 10(1–2): 88–110. <http://dx.doi.org/10.1080/19336918.2015.1120397>.
- Soncin, Francesca et al. 2018. "Comparative Analysis of Mouse and Human Placentae across Gestation Reveals Species-Specific Regulators of Placental Development." *Development (Cambridge)* 145(2).
- Soncin, Francesca, David Natale, and Mana M. Parast. 2015. "Signaling Pathways in Mouse and Human Trophoblast Differentiation: A Comparative Review." *Cellular and Molecular Life Sciences* 72(7): 1291–1302.
- Spies, Daniel, Peter F. Renz, Tobias A. Beyer, and Constance Ciaudo. 2019. "Comparative Analysis of Differential Gene Expression Tools for RNA Sequencing Time Course Data." *Briefings in Bioinformatics* 20(1): 288. /pmc/articles/PMC6357553/ (July 18, 2022).
- Sun, Huifang et al. 2006. "An E2F Binding-Deficient Rb1 Protein Partially Rescues Developmental Defects Associated with Rb1 Nullizygoty." *Molecular and Cellular Biology*

- 26(4): 1527. /pmc/articles/PMC1367194/ (February 6, 2022).
- Sutton, Gavin J. et al. 2022. "Comprehensive Evaluation of Deconvolution Methods for Human Brain Gene Expression." *Nature Communications* 2022 13:1 13(1): 1–18.
<https://www.nature.com/articles/s41467-022-28655-4> (July 27, 2022).
- Varberg, Kaela M. et al. 2021. "ASCL2 Reciprocally Controls Key Trophoblast Lineage Decisions during Hemochorial Placenta Development." *Proceedings of the National Academy of Sciences of the United States of America* 118(10).
<https://www.pnas.org/content/118/10/e2016517118> (December 21, 2021).
- Wang, Anyou, S. Claiborne Johnston, Joyce Chou, and Deborah Dean. 2010. "A Systemic Network for Chlamydia Pneumoniae Entry into Human Cells." *Journal of Bacteriology* 192(11): 2809–15. <https://journals.asm.org/doi/10.1128/JB.01462-09> (July 27, 2022).
- Wang, Anyou, Li Ren, and Hong Li. 2011. "A Systemic Network Triggered by Human Cytomegalovirus Entry." *Advances in Virology* 2011.
- Wang, Niya et al. 2016. "Mathematical Modelling of Transcriptional Heterogeneity Identifies Novel Markers and Subpopulations in Complex Tissues." *Scientific Reports* 2016 6:1 6(1): 1–12. <https://www.nature.com/articles/srep18909> (July 27, 2022).
- Watson, Erica D., and James C. Cross. 2005. "Development of Structures and Transport Functions in the Mouse Placenta." *Physiology* 20(3): 180–93.
- Woods, Laura, Vicente Perez-garcia, and Myriam Hemberger. 2018. "Regulation of Placental Development and Its Impact on Fetal Growth — New Insights From Mouse Models." *Frontiers in Endocrinology* 9(September): 1–18.
- Zhong, Yi et al. 2013. "Digital Sorting of Complex Tissues for Cell Type-Specific Gene Expression Profiles." *BMC Bioinformatics* 14(1): 1–10.
<https://bmcbioinformatics.biomedcentral.com/articles/10.1186/1471-2105-14-89> (July 27, 2022).

November 17, 2022

RE: Life Science Alliance Manuscript #LSA-2022-01788

Geetu Tuteja
Iowa State University
2106 Molecular Biology Building
2437 Pammel Drive
Ames, IA 50011-2140

Dear Dr. Tuteja,

Thank you for submitting your revised manuscript entitled "Identifying novel regulators of placental development using time series transcriptomic data and network analyses". We would be happy to publish your paper in Life Science Alliance pending final revisions necessary to meet our formatting guidelines.

- please address Reviewer 2's final comment
- please upload both your main and supplementary figures as single files and add a separate figure legend section to your main manuscript text
- please add a Running Title, Alternate Abstract/Summary Blurb, and a category for your manuscript to our system
- please add the Twitter handle of your host institute/organization as well as your own or/and one of the authors in our system
- please use the [10 author names, et al.] format in your references (i.e. limit the author names to the first 10)
- the data uploaded to GEO should now be made publicly accessible

Figure Check:

- please add sizes next to all blots
- please add scale bars to Figure 4B and S7C, and indicate their size in the legend

A. FINAL FILES:

B. MANUSCRIPT ORGANIZATION AND FORMATTING:

Sincerely,

Reviewer #1 (Comments to the Authors (Required)):

I previously reviewed this manuscript (Reviewer 1). I find that the reviewers have addressed all my previous questions and concerns. I also suggest that they have done an excellent job addressing the other reviewers.

Reviewer #2 (Comments to the Authors (Required)):

The authors have addressed all reviewers' points. Only one minor point needs to be addressed:
In multiple places (in both the manuscript and the rebuttal), the authors reference Soncin et al.'s 2015 review, where they mean to reference Soncin et al.'s 2018 manuscript (which has the microarray data). Please correct references to these two distinct studies throughout the manuscript.

Reviewer #3 (Comments to the Authors (Required)):

I am overall satisfied with the responses brought by the authors to my concerns.

November 22, 2022

RE: Life Science Alliance Manuscript #LSA-2022-01788R

Dr. Geetu Tuteja
Iowa State University
2106 Molecular Biology Building
2437 Pammel Drive
Ames, IA 50011-2140

Dear Dr. Tuteja,

Thank you for submitting your Research Article entitled "Identifying novel regulators of placental development using time-series transcriptome data". It is a pleasure to let you know that your manuscript is now accepted for publication in Life Science Alliance. Congratulations on this interesting work.

DISTRIBUTION OF MATERIALS:

Again, congratulations on a very nice paper. I hope you found the review process to be constructive and are pleased with how the manuscript was handled editorially. We look forward to future exciting submissions from your lab.

Sincerely,
